# Novel autophagy inducers by accelerating lysosomal clustering against Parkinson's disease

Yuki Date[1,2], Yukiko Sasazawa[2,3,4], Mitsuhiro Kitagawa[2], Kentaro Gejima[2], Ayami Suzuki[2], Hideyuki Saya[5,6], Yasuyuki Kida[7], Masaya Imoto[4], Eisuke Itakura[8], Nobutaka Hattori[2,3,4,9]*, Shinji Saiki[2,4,10]*

[1]Department of Biology, Graduate School of Science and Engineering, Chiba University, Inage-ku, Chiba, Japan; [2]Department of Neurology, Juntendo University Faculty of Medicine, Tokyo, Japan; [3]Research Institute for Diseases of Old Age, Juntendo University Graduate School of Medicine, Tokyo, Japan; [4]Division for Development of Autophagy Modulating Drugs, Juntendo University Faculty of Medicine, Tokyo, Japan; [5]Division of Gene Regulation, Institute for Advanced Medical Research, School of Medicine, Keio University, Tokyo, Japan; [6]Division of Gene Regulation, Cancer Center, Fujita Health University, Toyoake, Japan; [7]Biotechnology Research Institute for Drug Discovery, National Institute of Advanced Industrial Science and Technology (AIST), Tsukuba, Japan; [8]Department of Biology, Graduate School of Science, Chiba University, Inage-ku, Chiba, Japan; [9]Neurodegenerative Disorders Collaborative Laboratory, RIKEN Center for Brain Science, Saitama, Japan; [10]Department of Neurology, Institute of Medicine, University of Tsukuba, Ibaraki, Japan

*For correspondence:
nhattori@juntendo.ac.jp (NH);
ssaiki@md.tsukuba.ac.jp (SS)

Competing interest: The authors declare that no competing interests exist.

**Abstract** The autophagy-lysosome pathway plays an indispensable role in the protein quality control by degrading abnormal organelles and proteins including α-synuclein (αSyn) associated with the pathogenesis of Parkinson's disease (PD). However, the activation of this pathway is mainly by targeting lysosomal enzymic activity. Here, we focused on the autophagosome-lysosome fusion process around the microtubule-organizing center (MTOC) regulated by lysosomal positioning. Through high-throughput chemical screening, we identified 6 out of 1200 clinically approved drugs enabling the lysosomes to accumulate around the MTOC with autophagy flux enhancement. We further demonstrated that these compounds induce the lysosomal clustering through a JIP4-TRPML1-dependent mechanism. Among them, the lysosomal-clustering compound albendazole promoted the autophagy-dependent degradation of Triton-X-insoluble, proteasome inhibitor-induced aggregates. In a cellular PD model, albendazole boosted insoluble αSyn degradation. Our results revealed that lysosomal clustering can facilitate the breakdown of protein aggregates, suggesting that lysosome-clustering compounds may offer a promising therapeutic strategy against neurodegenerative diseases characterized by the presence of aggregate-prone proteins.

## Editor's evaluation

This study reports a valuable finding regarding the therapeutic efficacy of compounds fostering lysosomal clustering to enhance the clearance of protein aggregates in neurodegenerative disorders. The data were collected and analyzed using solid and validated methodology. While a deeper mechanistic understanding would have strengthened the study, the work will be of interest to cell

biologist work on autophagy and lysosome and medical biologists working on neurodegenerative diseases.

## Introduction

PD, the second most prevalent progressive neurodegenerative disorder after Alzheimer's disease, is characterized by dopamine depletion due to the loss of dopaminergic neurons in the substantia nigra pars compacta. PD affects a broad range of motor and nonmotor functions and develops in approximately 1% of individuals over the age of 65. One primary therapy for PD is dopamine supplementation, often through medications like levodopa. While levodopa has a transformative effect, its chronic use can result in adverse effects, such as wearing off and dyskinesia. Since current treatments do not prevent the loss of dopaminergic neurons, new therapeutic strategies are urgently needed to address the challenges of PD.

The aggregation and deposition of the αSyn protein, also known as Lewy bodies, within dopaminergic neurons in the substantia nigra play a critical role in the etiology of PD (*Spillantini et al., 1997*; *Postuma et al., 2015*). Research interest is increasing in the induction of autophagy, a primary protein degradation system that removes αSyn aggregates, using small-molecule compounds as a therapeutic approach (*Webb et al., 2003*; *Gao et al., 2019*; *Choi et al., 2020b*). During autophagy, an isolation membrane engulfs part of the cytoplasm, forming a double-membraned vesicle known as an autophagosome. This organelle then fuses with lysosomes, creating autolysosomes where lysosomal hydrolases break down the vesicle's contents. Notably, rapamycin, a recognized autophagy inducer, can inhibit αSyn aggregation in vivo and improve motor functions (*Crews et al., 2010*; *Dehay et al., 2010*). However, even with the discovery of autophagy inducers, no fully effective treatment has been established. For instance, while nilotinib boosts autophagic αSyn clearance in αSyn transgenic PD model mice, its performance in clinical trials was underwhelming. Therefore, compounds that can dismantle aggregated proteins using innovative mechanisms need to be developed.

Recent studies have highlighted the importance of lysosomal distribution in regulating autophagy and its associated genes. Lysosomes clustering around the (MTOC) are pivotal in managing autophagic flux. The close physical association between these clustered lysosomes and autophagosomes facilitates their fusion as well as inhibition of the mechanistic target of rapamycin complex1 (*Kimura et al., 2008*; *Korolchuk et al., 2011*). The transport of lysosomes to the MTOC is dependent on the dynein/dynactin complex and involves several protein pathways: (i) the small GTPase Rab7–Rab7 effector Rab-interacting lysosomal protein (RILP) pathway *Johansson et al., 2007*; *Rocha et al., 2009*; (ii) the transient receptor potential mucolipin 1 (TRPML1)–apoptosis-linked gene 2 (ALG2) pathway *Li et al., 2016*; and (iii) the lysosomal membrane protein TMEM55B-JNK-interacting protein 4 (JIP4) pathway (*Willett et al., 2017*). Furthermore, we recently identified an additional pathway involving the phosphorylated JIP4–TRPML1–ALG2 complex induced by oxidative stress (*Sasazawa et al., 2022*). We also hypothesize that the phosphorylation status of JIP4 at T217 by CaMK2G serves as a controlling switch, altering its binding affinity and thus dictating the pathway activated by JIP4.

Notably, disruptions in lysosome and autophagosome transport can contribute to neurodegenerative diseases. For example, dynactin mutations are linked to Perry's Disease, a rare hereditary neurodegenerative condition marked by autosomal dominant parkinsonism (*Konno et al., 2017*). Such mutations disrupt lysosomal distribution, leading to impaired autophagy and cell death (*Ishikawa et al., 2014*). Additionally, leucine-rich repeat kinase 2 (LRRK2), a causative protein in PD, is essential in regulating autophagosome and lysosomal trafficking, particularly in partnership with JIP4 (*Bonet-Ponce et al., 2020*; *Kluss et al., 2022a*; *Kluss et al., 2022b*).

Conversely, mounting evidence suggests that protein aggregates, including those causing neurodegenerative diseases, accumulate around MTOCs and are degraded via the autophagy-lysosome pathway. Proteasome inhibitors, for example, aggregate near the nucleus and undergo autophagic degradation (*Jänen et al., 2010*; *Choi et al., 2020a*). In the same vein, the pathogenic protein mutant huntingtin, linked to Huntington's disease, forms aggresomes near the nucleus and is degraded by autophagy (*Waelter et al., 2001*; *Ma et al., 2022*). Similarly, the αSyn aggregates that form Lewy bodies in patients with PD accumulate near the MTOC (*Olanow et al., 2004*), hinting at the possibility that MTOC-centric autophagy facilitates their removal. Prior research has shown that Arl8b

overexpression inhibits the degradation of αSyn-A53T (*Korolchuk et al., 2011*), suggesting that lysosomal retrograde trafficking is vital in the breakdown of αSyn-A53T.

Therefore, we hypothesize that compounds that induce lysosomal clustering near MTOCs could minimize the distance between lysosomes, autophagosomes, and degradation substrates, thereby enhancing the degradation of protein aggregates. Guided by this concept, we developed a screening system that employed high-content image screening to identify a novel set of autophagy inducers that encourage lysosomal clustering. We further demonstrated that these compounds effectively degrade αSyn aggregates, underscoring the potential of enhancing lysosomal clustering as a strategy for eliminating protein aggregates.

## Results

### Cell-based screening for compounds inducing lysosomal clustering

To identify compounds that enhance the degradation of αSyn aggregates, we first screened for the ability to induce lysosomal clustering and then assessed their autophagy-inducing activity (*Figure 1A*).

For the identification process, we devised a high-content image-screening system using INCell Analyzer2200. As a starting point, we engineered SH-SY5Y cells to stably express both LGP120-mCherry (a lysosomal marker) and GFP-γ-tubulin (an MTOC marker) (*Figure 1B*). Using the INCellAnalyzer2200, we captured fluorescence images of GFP and mCherry. The distribution of lysosomes was then quantified using ImageJ software. We determined the MTOC location using the GFP-γ-tubulin spot signal, then enclosed the MTOC in a circle roughly 7 μm in diameter. The intensity of the LGP120-mCherry signal within this circle was gauged, and its ratio to the overall intensity for the cell was determined (*Figure 1C*). This ratio, when related to the control, was defined as the lysosomal clustering value.

To verify the accuracy of our approach, we cultivated the cells under starvation conditions, using this as the positive control, and subsequently assessed lysosomal distribution. Starvation resulted in evident lysosomal clustering, noticeable in fluorescence images and corroborated by the elevated lysosomal clustering value (*Figure 1D and E*), thereby underscoring the utility of our high-content image-screening system.

Using this screening method, we assessed approximately 1200 compounds, all of which are clinically approved in Japan. Of these, 63 exhibited a high lysosomal clustering value greater than 1.1 (*Figure 1F* and *Figure 1—source data 1*).

### Topoisomerase inhibitor and benzimidazole-induced lysosomal clustering and autophagy

To identify the autophagy inducers, we assessed the autophagic activity of the selected compounds using RFP–GFP tandem fluorescent-tagged LC3 (R-G-LC3) as a second screening test (*Figure 2A*; *Takayama et al., 2017*). We established SH-SY5Y cells that stably expressed R-G-LC3. During the fusion of autophagosomes with mammalian lysosomes in these cells, GFP fluorescence is quenched due to its pH sensitivity and subsequently degraded by lysosomal proteases. In contrast, RFP is resistant to both acidic conditions and lysosomal proteases, resulting in accumulation in lysosomes. The distinct properties of RFP and GFP enabled us to evaluate autophagic activity precisely by measuring the RFP/GFP fluorescence ratio in a cell using flow cytometry (*Figure 2—figure supplement 1*).

From the secondary screening of the 63 compounds identified in the first screening, we identified 15 compounds that induced autophagy with lysosomal clustering (*Figure 2A* and *Figure 1—source data 1*). Because five compounds are auto-fluorescent chemicals (denoted by gray dots in *Figure 2A*), we further evaluated the effects of the remaining 10 compounds on endogenous lysosomal clustering. We identified six compounds that induced lysosomal clustering and subsequent enhancement of autophagy (*Figure 2B*), which were categorized into two types: topoisomerase II inhibitors (topo-i), including teniposide, amsacrine, and etoposide; and benzimidazole class anthelmintics, comprising albendazole, oxibendazole, and mebendazole. An immunofluorescence assay of R-G-LC3-expressing SH-SY5Y cells revealed that both starvation and treatment with either teniposide or albendazole promoted autophagosomal perinuclear clustering and increased their colocalization with lysosomes (autolysosomes) (*Figure 2—figure supplement 2A*).

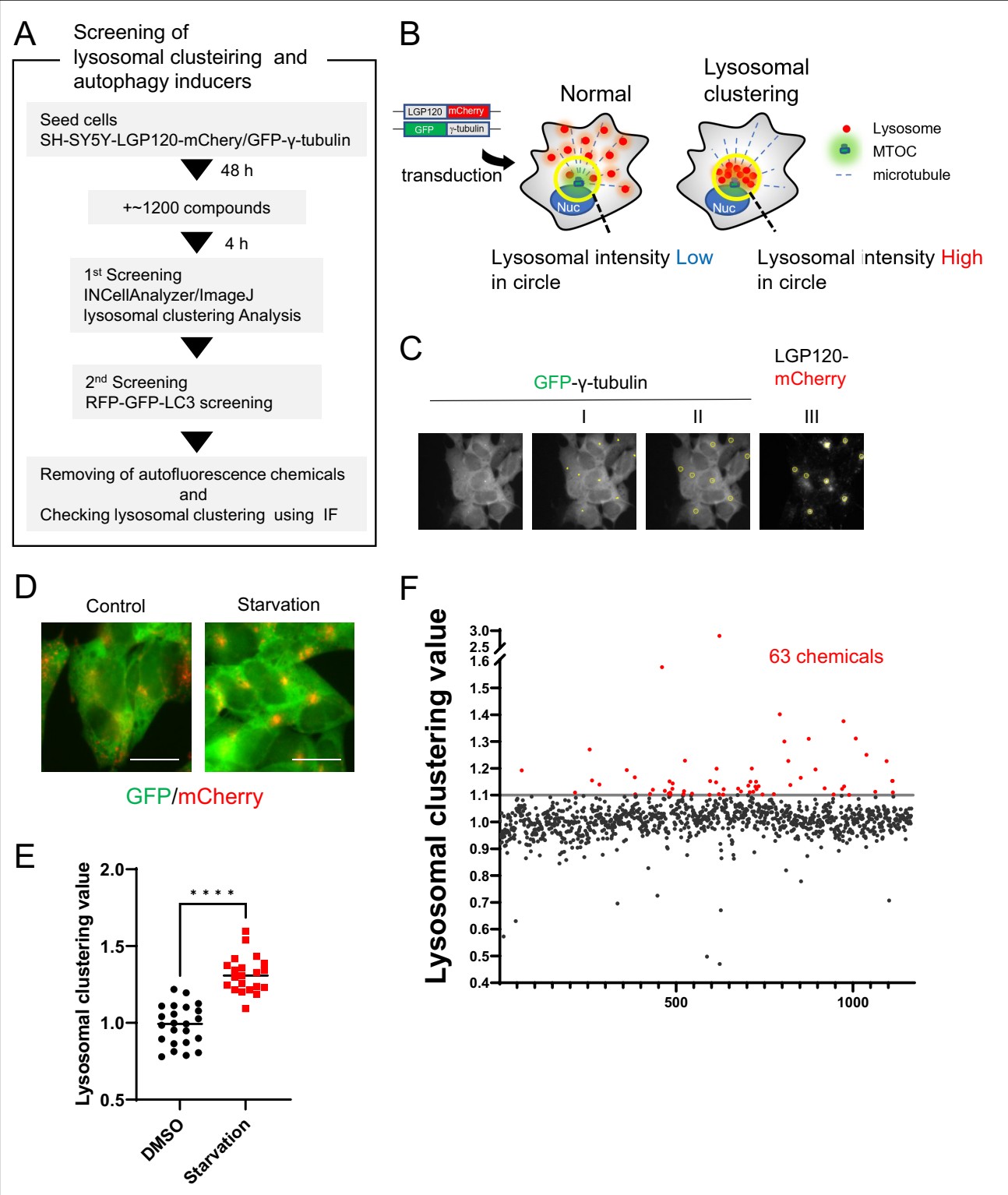

**Figure 1.** Method for screening lysosomal-clustering compounds. (**A**) Scheme for the process of screening for lysosomal-clustering compounds. (**B**) Strategies for screening lysosomal-clustering compounds. If lysosomes accumulate around the microtubule-organizing center (MTOC), the fluorescent intensity (red) in the circle increases. (**C**) INCellAnalyzer2200 images of SH-SY5Y cells co-expressing GFP-γ-tubulin and LGP120-mCherry. I–III details the procedure for lysosomal-clustering analysis using ImageJ. (**I**) Image depicting GFP-γ-tubulin fluorescence. The MTOC position as extracted using ImageJ processing. (**II**) A circle approximately 7 μm in diameter is placed at the central coordinates of the MTOC position. (**III**) The circle is reflected onto the LGP120 image to measure the LGP120 fluorescence intensity (lysosomal clustering value) within the circle. (**D**) SH-SY5Y cell lines

*Figure 1 continued on next page*

*Figure 1 continued*

co-expressing GFP-γ-tubulin and LGP120-mCherry were treated with either control medium or starvation medium (positive control). The results were analyzed using an INCellAnalyzer2200 and ImageJ. Scale bar: 20 μm. (**E**) Graph plotting the lysosomal clustering values (n>20). ****p<0.0001, Wilcoxon test. (**F**) Graph presenting the fold changes in the lysosome-clustering value for 1200 chemicals relative to the control. Chemicals with over a 1.1-fold change in lysosome-clustering value relative to the control were identified as lysosome-clustering compounds.

The online version of this article includes the following source data for figure 1:

**Source data 1.** Data table of lysosomal clustering chemical screening.

Furthermore, an autophagy flux assay conducted via western blot indicated that these compounds exhibited LC3B lipidation, an effect that was amplified by the lysosomal inhibitor bafilomycin A1 (*Mizushima and Yoshimori, 2007*; *Figure 2C*). The results indicate that we successfully identified a novel class of six autophagy inducers that promoted lysosomal clustering.

## Six compounds induced lysosomal clustering and autophagy independently of mTORC1

Previous studies have shown that lysosomal retrograde transport regulates autophagic flux by facilitating autophagosome formation by suppressing mTORC1 and expediting fusion between autophagosomes and lysosomes (*Kimura et al., 2008*; *Korolchuk et al., 2011*). Conversely, we recently found that acrolein/$H_2O_2$ induces lysosomal clustering in an mTOR-independent manner (*Sasazawa et al., 2022*). In this study, we aimed to identify pharmacologic agents that act downstream rather than upstream in the autophagy pathway, with the goal of minimizing side effects. Therefore, we evaluated the effects of the compounds on the mTOR pathway. We examined whether the six compounds modulated mTORC1 activity by regulating lysosomal positioning. This was accomplished by monitoring the phosphorylation states of mTOR and the mTORC1 downstream effectors, such as ribosomal S6 protein kinase (p70S6K), ribosomal S6 protein (S6), and ULK1 (*Klionsky et al., 2016*). The addition of Torin1, a selective mTOR inhibitor, in the starvation medium inhibited the phosphorylation of S6, p70S6K, mTOR, and ULK1. In contrast, the six compounds only marginally reduced the p-p70S6K levels without affecting p-S6, p-mTOR, and p-ULK1 (*Figure 3A*). Given the lack of a clear link between lysosomal accumulation and mTORC1 activity, as demonstrated by Torin1 treatment (*Figure 3B*, *Figure 2—figure supplement 2A and B*), the six compounds could induce autophagy independently of mTORC1 activity, indicating their potential as promising candidates for PD therapy.

## Topoisomerase inhibitor and benzimidazole transported lysosomes dependently through the JIP4–TRPML1 pathway

Various factors regulate lysosomal clustering. Four primary dynein-mediated pathways for lysosomal clustering have been identified: Rab7–RILP (*Johansson et al., 2007*), TRPML1–ALG2 (*Li et al., 2016*), TMEM55B–JIP4 (*Willett et al., 2017*), and TRPML1–JIP4–ALG2 (*Sasazawa et al., 2022*) pathways. To discern which pathways are involved in the lysosomal retrograde transport initiated by the selected compounds, we first examined the effects of *Rab7*, *RILP*, *ALG2*, *TRPML1*, *TMEM55B*, and *JIP4* knockdown on the lysosomal distribution induced by teniposide and albendazole, which typify topo-i and benzimidazole class anthelmintics, respectively. The knockdown efficiency against these genes was validated via western blotting and quantitative reverse-transcription polymerase chain reaction (*Figure 4—figure supplement 1*). *JIP4* and *TRPML1* knockdown hindered teniposide-induced lysosomal clustering, whereas *TMEM55B*, *ALG2*, *Rab7*, and *RILP* knockdown did not. Conversely, albendazole-induced lysosomal clustering was hindered by *JIP4*, *TRPML1*, *ALG2*, and *Rab7* knockdown, but *TMEM55B* and *RILP* knockdown had no such effect (*Figure 4A and B*). Moreover, lysosomal accumulation induced by teniposide/albendazole treatment was not attenuated by siRNA knockdown against *TMEM55B*, which is key in inducing starvation-dependent lysosomal clustering around the MTOC (*Figure 4—figure supplement 2A*; *Willett et al., 2017*). Therefore, the knockdown efficiency of *TMEM55B* was sufficient, and these compounds (topo-i and benzimidazole) induced lysosomal clustering independently of TMEM55B, unlike when under starvation conditions.

Conversely, the enhancement of lysosomal retrograde transport in *RILP* knockdown cells in *Figure 4B* suggests the potential involvement of RILP in anterograde transport. However, to the best

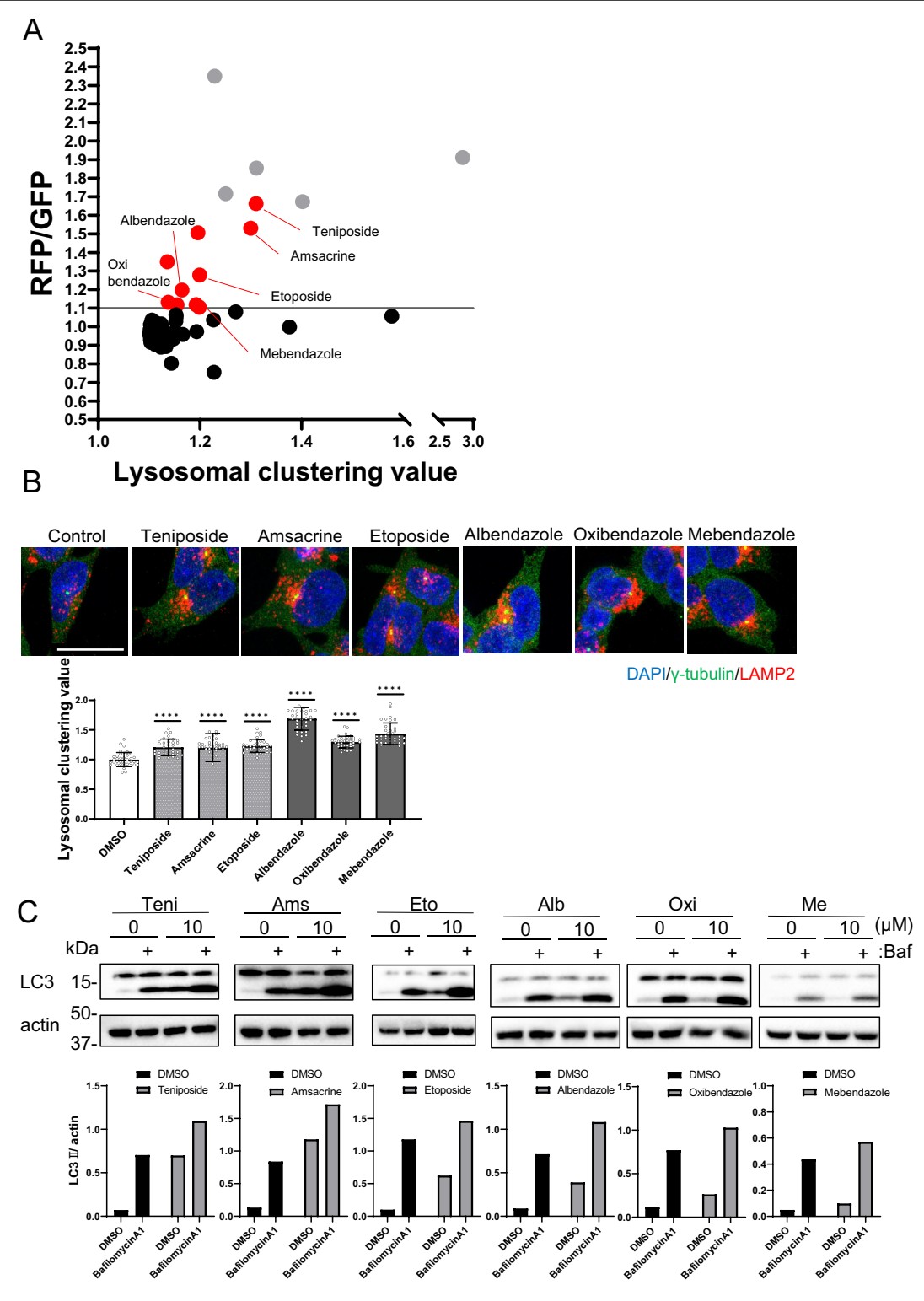

**Figure 2.** Identification of autophagy inducers via lysosomal clustering. (**A**) SH-SY5Y cells expressing red fluorescent protein (RFP)-green fluorescent protein (GFP) tandem fluorescent-tagged LC3 (**R–G–LC3**) were analyzed by flow cytometer after treatment with 63 lysosomal-clustering compounds for 20 hr. The graph illustrates the fold change in the RFP/GFP ratio for these 63 chemicals relative to the control. Auto-fluorescent chemicals were excluded if they had >1.1 fold change in RFP/GFP ratio relative to the control. After confirming endogenous lysosome accumulation using immunofluorescence, six lysosomal-clustering autophagy inducers were ultimately identified. Gray dots represent autofluorescence chemical data. (**B**) SH-SY5Y cells were treated with teniposide (10 μM), amsacrine (10 μM), etoposide (10 μM), albendazole (10 μM), oxibendazole (1 μM), and mebendazole (5 μM) for

*Figure 2 continued on next page*

*Figure 2 continued*

4 hr. Cells were then fixed and stained with the indicated antibody and DAPI. Images were captured using a confocal microscope. Scale bar: 20 μm. Under similar conditions, these cells were analyzed using an INCellAnalyzer2200 and ImageJ for lysosomal clustering. The graph shows the lysosomal clustering values (n>30, from three biological replicates). Data are expressed as mean ± standard deviation. ****p<0.0001, ***p<0.001, **p<0.01, *p<0.05, two-way analysis of variance and Tukey's test. N.S., not statistically significant. (**C**) SH-SY5Y cells underwent treatment with lysosome-clustering compounds for 4 hr. Cell lysates were immunoblotted with the indicated antibodies. The amount of LC3II was estimated using ImageJ software (bottom panel).

The online version of this article includes the following source data and figure supplement(s) for figure 2:

**Source data 1.** Uncropped blot images of Figure 2C.

**Figure supplement 1.** Flow cytometer analysis of red fluorescent protein (RFP)-green fluorescent protein (GFP) tandem fluorescent-tagged LC3 (**R–G–LC3**) during second screening.

**Figure supplement 2.** Confocal images of SH-SY5Y expressing red fluorescent protein (RFP)-green fluorescent protein (GFP) tandem fluorescent-tagged LC3 (RFP-GFP-LC3).

of our knowledge, no reports have investigated this matter. We presume that negative feedback mechanisms may be present.

These observations suggest that topo-i-driven lysosomal clustering is mediated by JIP4 and TRPML1, whereas albendazole-induced clustering involves JIP4, TRPML1, ALG2, and Rab7. Furthermore, in JIP4 knockout (KO) cells, lysosomal clustering induction by all six compounds was suppressed

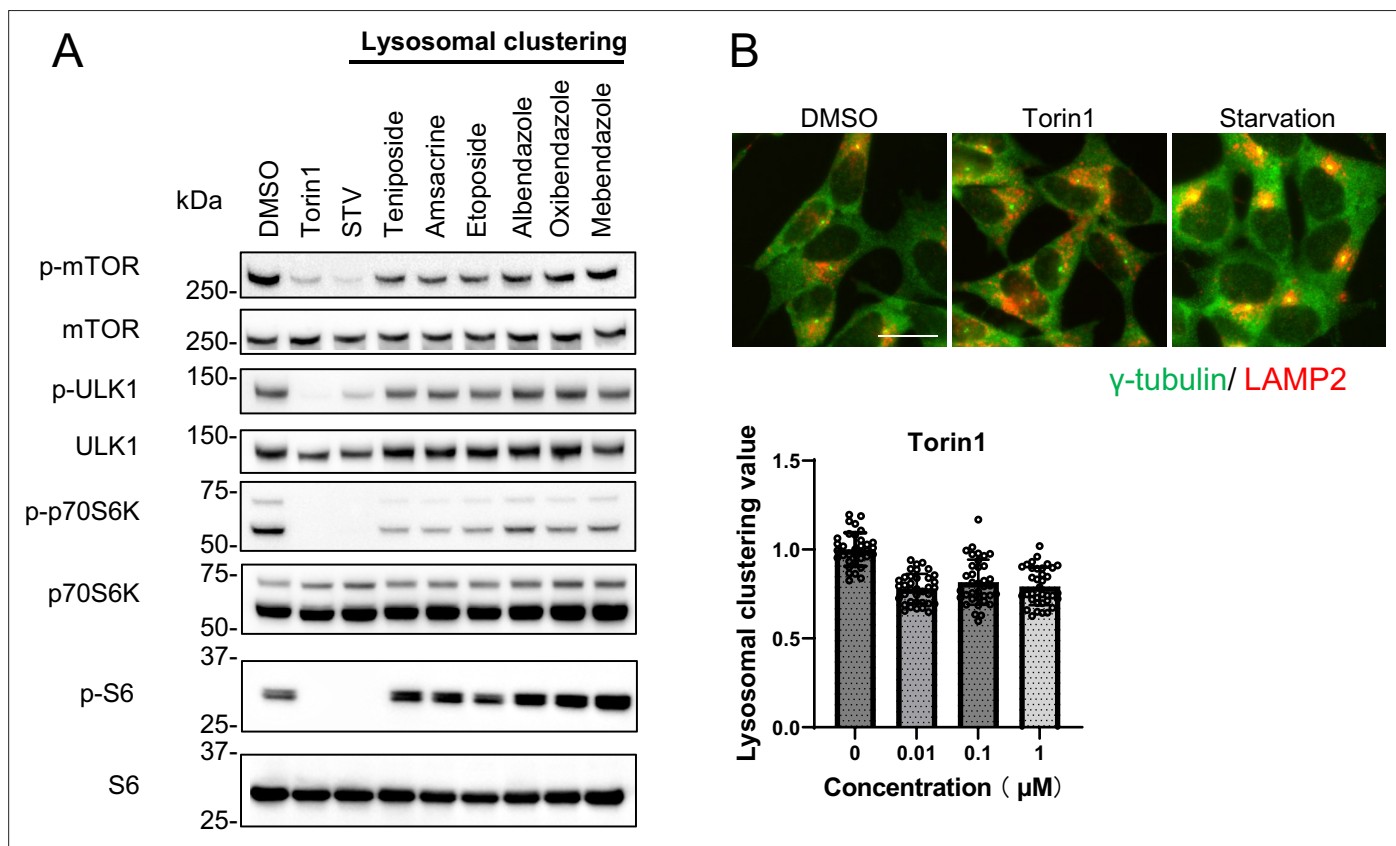

**Figure 3.** Lysosomal-clustering compounds are not dependent on mTORC1 activity. (**A**) SH-SY5Y cells were treated with starvation medium, Torin1 (1 μM), teniposide (10 μM), amsacrine (10 μM), etoposide (10 μM), albendazole (10 μM), oxibendazole (1 μM), or mebendazole (5 μM) for 4 hr. Cell lysates were then immunoblotted with the specified antibody. (**B**) SH-SY5Y cells treated with Torin1 (1 μM) for 4 hr. These cells were fixed, stained with anti-γ-tubulin (green) and anti-LAMP2 (red) antibodies, and imaged using an INCellAnalyzer2200. Scale bar: 20 μm. INCellAnalyzer2200 images were then analyzed using ImageJ for lysosomal clustering. The graph presents the lysosomal clustering values (n>30). Data are expressed as mean ± standard deviation.

The online version of this article includes the following source data for figure 3:

**Source data 1.** Uncropped blot images of Figure 3A.

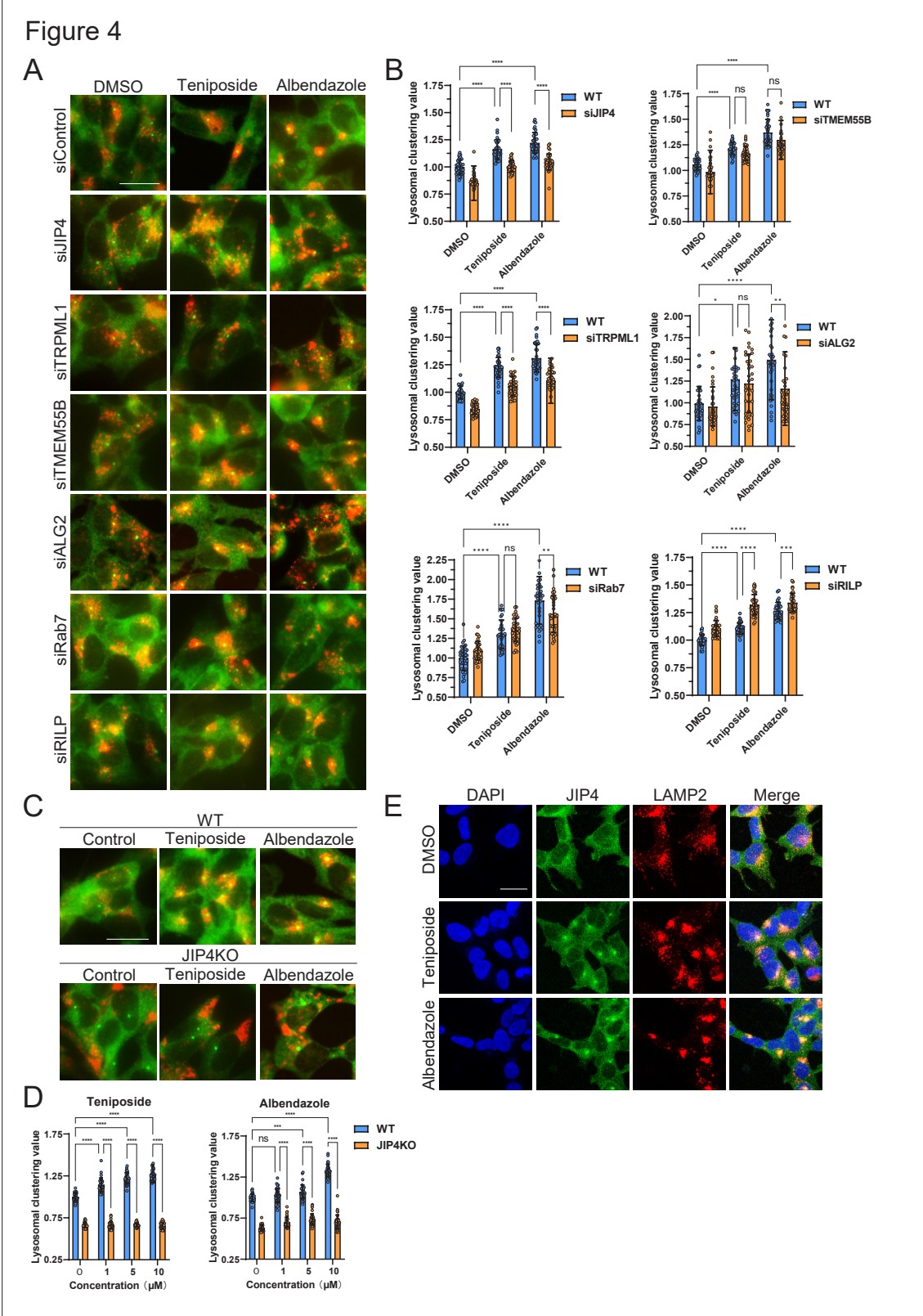

**Figure 4.** Lysosome-clustering compounds accumulate lysosomes in microtubule-organizing center (MTOC) in a JIP4-dependent manner. (**A**) SH-SY5Y cells were transfected with the indicated siRNAs for 48 hr and then treated with teniposide (10 µM) and albendazole (10 µM) for 4 hr. Cells were fixed and stained with anti-γ-tubulin (green) and anti-LAMP2 (red) antibodies. Images were captured using an INCellAnalyzer2200. Scale bar: 20 µm. (**B**) INCellAnalyzer2200 images were analyzed for lysosomal clustering using ImageJ. The graph presents the lysosomal clustering values (n>30, from

*Figure 4 continued on next page*

*Figure 4 continued*

three biological replicates). Data are expressed as mean ± standard deviation. ****p<0.0001, ***p<0.001, **p<0.01, *p<0.05, two-way analysis of variance (ANOVA) and Tukey's test. N.S., not statistically significant. The experiment was technically replicated at least three times. (**C**) SH-SY5Y WT and JIP4 knockout cells were cultured in 96-well black plates and treated with teniposide (1, 5, or 10 µM) and albendazole (1, 5, or 10 µM). Cells were fixed and stained with anti-γ-tubulin (green) and anti-LAMP2 (red) antibodies. Images were captured using an INCellAnalyzer2200. (**D**) INCellAnalyzer2200 images were analyzed using ImageJ for lysosomal clustering. The graph presents the lysosomal clustering values for wild-type or JIP4 knockout cells (n>30, from three biological replicates). Data are expressed as mean ± standard deviation. ****p<0.0001, ***p<0.001, **p<0.01, *p<0.05, two-way ANOVA and Tukey's test. N.S., not statistically significant. The experiment was technically replicated at least three times. (**E**) SH-SY5Y cells were treated with teniposide (10 µM) and albendazole (10 µM) for 4 hr. Cells were fixed and stained with anti-JIP4 (green) and anti-LAMP2 (red) antibodies and 4',6-diamidino-2-phenylindole.

The online version of this article includes the following figure supplement(s) for figure 4:

**Figure supplement 1.** siRNAs and knockdown efficiency of each lysosomal factor.

**Figure supplement 2.** Lysosomal-clustering analysis of other lysosomal-clustering compounds using JIP4KO cells.

(*Figure 4C and D*, and *Figure 4—figure supplement 2B*). Immunostaining using JIP4 antibodies revealed increased colocalization of JIP4 with LAMP2 under both teniposide and albendazole treatment conditions (*Figure 4E*). Collectively, these results show that, while teniposide and albendazole may stimulate lysosomal clustering via different pathways, both TRPML1 and JIP4 are instrumental in each pathway.

## Topoisomerase inhibitor regulated lysosomal transport via phosphorylated JIP4

We previously determined that JIP4 plays a crucial role in regulating lysosomal clustering triggered by both oxidative stress and starvation. Interestingly, these two processes employ distinct pathways influenced by the phosphorylation status of JIP4. Specifically, oxidative stress results in the phosphorylation of JIP4 at T217 through CaMK2G activation, regulating lysosomal retrograde transport in tandem with TRPML1 and ALG2. In contrast, starvation mediates lysosomal retrograde transport predominantly through the TRPML1–ALG2 and TMEM55B–JIP4 pathways without JIP4 phosphorylation (*Sasazawa et al., 2022*).

Given this background, we determined whether lysosomal clustering induced by topo-i and benzimidazole is dependent on JIP4 phosphorylation. We employed Jak3 inhibitor VI, previously identified as an inhibitor of the JIP4 kinase CaMK2G, which regulates oxidative stress-induced lysosomal clustering (*Sasazawa et al., 2022*). Notably, Jak3 inhibitor VI effectively suppressed lysosomal clustering induced by the three topo-i compounds. In contrast, the effects of benzimidazole remained largely unaffected (*Figure 5A and B*). Moreover, Phos-tag sodium dodecyl sulfate-polyacrylamide gel electrophoresis (SDS-PAGE) analyses revealed that all three topo-i compounds induced JIP4 phosphorylation, an effect counteracted by Jak3 inhibitor VI (*Figure 5C*). siCaMK2G also inhibited lysosomal clustering induced by all three topo-i compounds (*Figure 5D and E*). In addition, $Ca^{2+}$ imaging showed that teniposide, but not albendazole, upregulated $Ca^{2+}$ flux (*Figure 5—figure supplement 1*). This suggests that CaMK2G-phosphorylated JIP4 is pivotal for topo-i-induced lysosomal clustering. In addition, rescue experiments in JIP4 KO SH-SY5Y cells, which re-expressed either flag-tagged wild-type JIP4 or a phosphorylation-defective variant (T217A JIP4), revealed that only the cells with wild-type flag-JIP4 could recover the teniposide-induced lysosomal clustering phenotype (*Figure 5F and G*). This underscores the indispensability of JIP4 phosphorylation at T217 for the changes in lysosomal distribution induced by topo-i. In summary, while topo-i-induced lysosomal clustering is promoted by phosphorylated JIP4 (T217) in conjunction with TRPML1 (but not ALG2), benzimidazole-driven lysosomal retrograde transport, exemplified by albendazole, engages the TRPML1–JIP4–ALG2 and Rab7 pathways.

## Lysosomal-clustering compounds efficiently induced autophagy by transporting autophagosomes and lysosomes in a JIP4-dependent manner

Next, we confirmed whether lysosomal clustering is essential for the autophagy triggered by the lysosomal-clustering compounds. We utilized a quantifiable HaloTag-LC3B assay to measure

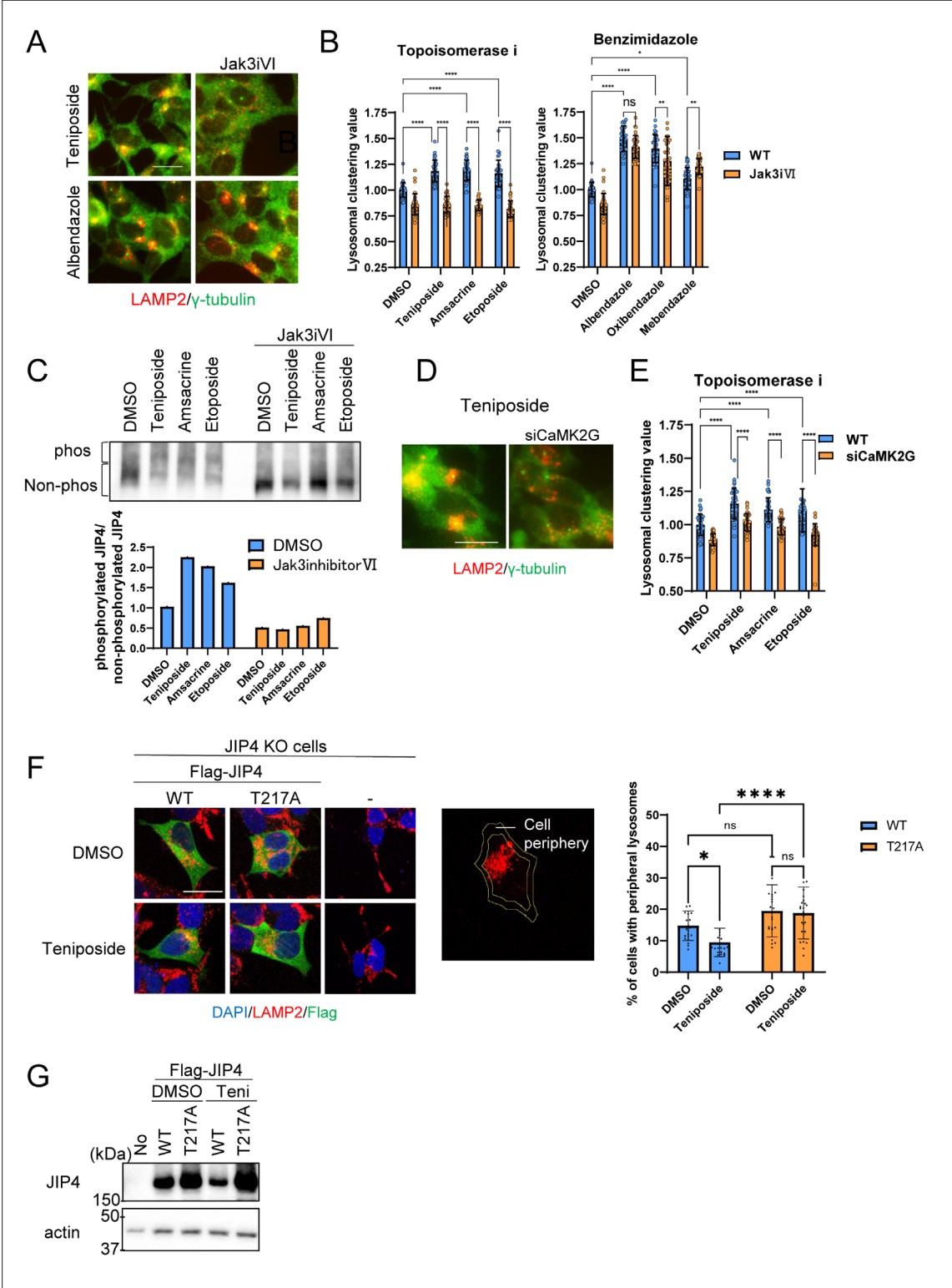

**Figure 5.** Lysosomal clustering induced by topoisomerase inhibitor requires JIP4 phosphorylation. (**A**) SH-SY5Y cells were treated with dimethyl sulfoxide (DMSO), teniposide (10 μM), amsacrine (10 μM), etoposide (10 μM), albendazole (10 μM), oxibendazole (1 μM), or mebendazole (5 μM) with or without Jak3 inhibitor VI for 4 hr. Cells were fixed and stained with anti-γ-tubulin (green) and anti-LAMP2 (red) antibodies. Images were captured using an INCellAnalyzer2200. Scale bar: 20 μm. (**B**) INCellAnalyzer2200 images were analyzed using ImageJ for lysosomal clustering. The graph presents the lysosomal clustering values for DMSO or Jak3 inhibitor VI (n>30, from 3 biological replicates). Data are expressed as mean ± standard deviation. ****p<0.0001, ***p<0.001, **p<0.01, *p<0.05, two-way analysis of variance (ANOVA) and Tukey's test. N.S., not statistically significant. The experiment

*Figure 5 continued on next page*

*Figure 5 continued*

was technically replicated at least three times. (**C**) SH-SY5Y cells were treated with teniposide (10 μM), amsacrine (10 μM), or etoposide (10 μM) with or without Jak3 inhibitor VI for 4 hr. Cell lysates underwent Phos-tag sodium dodecyl sulfate-polyacrylamide gel electrophoresis (SDS-PAGE) and were immunoblotted with an anti-JIP4 antibody. The graph displays the ratio of non-phosphorylated JIP4 to phosphorylated JIP4. (**D**) SH-SY5Y cells were transfected with CaMK2G siRNAs for 48 hr and then treated with teniposide (10 μM), amsacrine (10 μM), or etoposide (10 μM) for 4 hr. Cells were fixed and stained with anti-γ-tubulin (green) and anti-LAMP2 (red) antibodies. Images were captured using an INCellAnalyzer2200. Scale bar: 20 μm. (**E**) INCellAnalyzer2200 images were analyzed using ImageJ for lysosomal clustering. The graph presents the lysosomal clustering values (n>30, from three biological replicates). Data are expressed as mean ± standard deviation. ****p<0.0001, two-way ANOVA, and Tukey's test. The experiment was technically replicated at least three times. (**F**) JIP4 KO cells were transfected with flag-tagged JIP4 (wild-type [WT] and T217A) for 24 hr and treated with teniposide (10 μM) for 4 hr. Cells were fixed and stained with the indicated antibodies. Scale bar: 20 μm. Approximately 70% of the cell area toward the cell center was automatically delineated using ImageJ, with the region excluded from this defined as the cellular peripheral region. The graph displays the percentage of cells with peripheral lysosomes. Data are expressed as mean ± standard deviation. *p<0.05, ****p<0.0001, two-way ANOVA and Tukey's test (n>20, from three biological replicates). (**G**) JIP4 KO cells were transfected with flag-tagged JIP4 (wild-type and T217A) for 24 hr and treated with teniposide (10 μM) for 4 hr. Cell lysates underwent SDS-PAGE and were immunoblotted with anti-JIP4 and anti-actin antibodies.

The online version of this article includes the following source data and figure supplement(s) for figure 5:

**Source data 1.** Uncropped blot images of Figure 5C and 5G.

**Figure supplement 1.** Calcium flux analysis of lysosomal clustering chemicals.

autophagic flux (*Yim et al., 2022*). In cells expressing HaloTag-LC3B, the HaloTag degrades within lysosomes in the absence of ligands. However, when ligands are introduced, the HaloTag portion accumulates inside lysosomes. This accumulation serves as an estimate of autophagic activity (*Yim et al., 2022*). We expressed HaloTag-LC3B in both parental SH-SY5Y cells and JIP4 KO cells and evaluated autophagic flux by adding topo-i and benzimidazole. JIP4 KO cells displayed reduced accumulation of the cleaved Halo bands on the in-gel fluorescence images for both compounds (*Figure 6A and B*). These findings reveal that both topo-i and benzimidazole provoke JIP4-dependent lysosomal clustering, subsequently inducing autophagy. Interestingly, JIP4 expression levels decreased in response to lysosomal-clustering compounds (*Figure 6A and B*). Moreover, the decrease in JIP4 expression in response to teniposide was suppressed by co-treatment with bafilomycin A1 (*Figure 6—figure supplement 1*), indicating that JIP4 is eventually degraded by autophagy, although it is essential for lysosomal transport.

Furthermore, immunofluorescence assay demonstrated that starvation, teniposide, and albendazole treatments promoted autophagosomal perinuclear clustering and enhanced their colocalization with lysosomes to form autolysosomes (*Figure 6C*). Interestingly, in JIP4 KO cells, lysosomal clustering and the accumulation of autophagosomes to the MTOC were inhibited. The accumulation of autolysosomes (indicated by the colocalization of Halo-LC3 with lysosomes) was observed at the cell periphery following teniposide treatment (*Figure 6C*, enlarged image), suggesting the induction of autophagy. As demonstrated in our previous study (*Ishikawa et al., 2019*), the inhibition of retrograde vesicular transport (e.g. through dynactin knockdown) causes both autophagosomes and lysosomes to disperse to the cell periphery, resulting in residual autolysosome formation there and preserving some autophagic flux. This likely caused the slight yet notable decline in autophagic flux in JIP4 KO cells treated with topo-i and benzimidazole. Conversely, in the presence of albendazole or under starvation conditions in JIP4 KO cells, the accumulation of autolysosomes at the cell periphery was infrequently observed. This supports the presumption that lysosomal clustering induced by albendazole or starvation is also mediated by pathways other than JIP4, such as the Rab7-mediated pathway.

Furthermore, autophagosomes that did not colocalize with lysosomes were detected in the cytoplasm following treatments with teniposide and albendazole and under starvation conditions in JIP4 KO cells (*Figure 6C*, indicated by a white arrow). These findings imply that teniposide- and albendazole-induced autophagy stimulates JIP4-dependent lysosomal movement to induce autophagy and facilitate the formation of autolysosomes. Moreover, these results suggest the potential role of JIP4, not only in lysosomal retrograde transport, but also in autophagosomal retrograde transport. In summary, in JIP4 KO cells, lysosomes and autophagosomes are directed toward the cell periphery, thereby retaining some level of autophagic flux.

In addition, to assess the ability of lysosomal-clustering compounds to induce autophagy by promoting the formation of autolysosomes, we compared the autophagic activity between Halo-LC3 and co-treatment with Torin1, which does not induce lysosome clustering, and a lysosomal-clustering

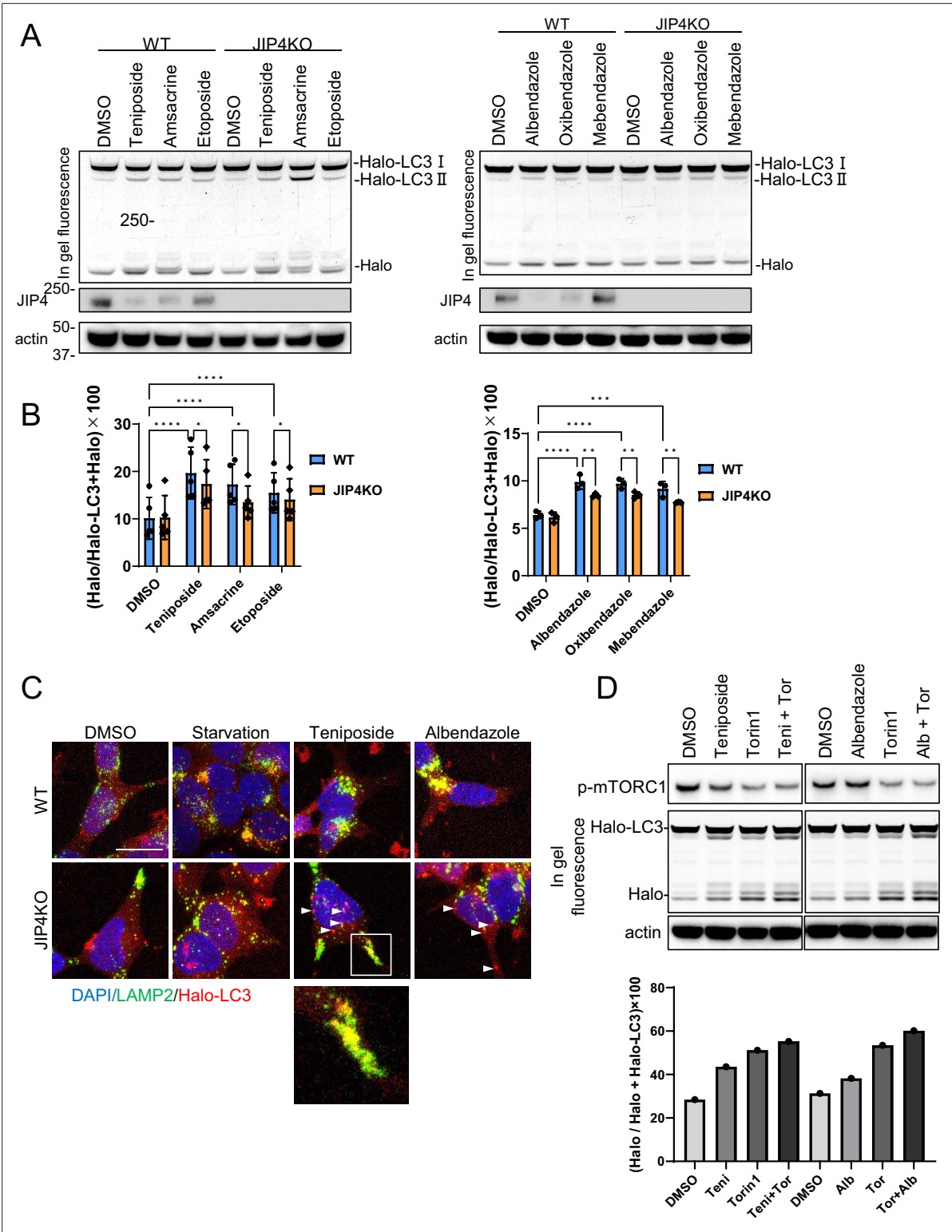

**Figure 6.** Lysosomal clustering dependent on JIP4 is slightly involved in autophagy activity. (**A**) SH-SY5Y cells stably expressing Halo-LC3 were labeled for 20 min with 100 nM tetramethylrhodamine (TMR)-conjugated ligand in a nutrient-rich medium. After washing with phosphate-buffered saline (PBS) and incubating the cells in normal medium for 30 min, cells were treated with dimethyl sulfoxide (DMSO), teniposide (10 μM), amsacrine (10 μM), etoposide (10 μM), albendazole (10 μM), oxibendazole (1 μM), or mebendazole (5 μM) for 4 hr. Cell lysates were immunoblotted with the indicated antibody and analyzed by in-gel fluorescence detection. (**B**) Quantification of results shown in panel A. The HaloTMR band intensity was normalized by the sum of the band intensities of HaloTMR-LC3B and HaloTMR. The vertical axis of the graph represents the intensity multiplied by 100. Mean values

*Figure 6 continued on next page*

*Figure 6 continued*

of data from five or three experiments are shown. Data are expressed as mean ± standard deviation. ****p< 0.0001, ***p<0.001, **p<0.01, *p<0.05, two-way analysis of variance and Tukey's test. (**C**) Cells treated as in panel A were fixed and stained with an anti-LAMP2 antibody (green). These cells were imaged using a confocal microscope. Scale bar: 20 µm. White arrows indicate Halo-LC3 dots. The magnified image shows LAMP2 (green) and Halo dots (red) accumulating at the cell periphery. (**D**) SH-SY5Y cells stably expressing Halo-LC3 were labeled for 20 min with 100 nM TMR-conjugated ligand in a nutrient-rich medium. After washing with PBS and incubating the cells in normal medium for 30 min, cells were treated with DMSO, teniposide (10 µM), albendazole (10µM), and/or Torin1 (100 nM) for 8 hr. Cell lysates were immunoblotted with the indicated antibody and analyzed by in-gel fluorescence detection (left). The HaloTMR band intensity was normalized by the sum of the band intensities of HaloTMR-LC3B and HaloTMR (right).

The online version of this article includes the following source data and figure supplement(s) for figure 6:

**Source data 1.** Uncropped gel fluorescence images of Figure 6A and 6D.

**Figure supplement 1.** Western blot analysis of JIP4 expression.

compound. Cotreatment with Torin1 and teniposide or albendazole-induced autophagy more effectively than Torin1 treatment alone, without affecting mTOR inhibition activity (*Figure 6D*). These findings indicate that the induction of autophagy by lysosome-clustering compounds is not caused by autophagosome formation but by the formation of autolysosomes.

## Albendazole-degraded proteasome inhibitor MG132-induced aggregates

Because topo-i exhibits cytotoxicity and is clinically used as an anti-cancer drug, we explored the ability of albendazole to induce protein aggregate degradation through autophagy. Previous reports have shown that aggregates caused by proteasomal inhibition can be degraded by autophagy (*Jänen et al., 2010*; *Choi et al., 2020a*), and the significance of lysosomal clustering has also been highlighted (*Zaarur et al., 2014*). MG132, a well-known proteasomal inhibitor, increased p62 and polyubiquitinated protein levels in the Triton-X insoluble fraction. This increase declined time-dependently after the washout of MG132 (*Figure 7A*). Furthermore, the reduction in p62 and ubiquitin levels following the MG132 washout was markedly suppressed in autophagy-deficient FIP200 KO cells (*Figure 7B*). These results show that basal autophagy degrades the MG132-induced protein aggregates. Notably, albendazole treatment in conjunction with the MG132 washout induced p62 reduction more rapidly than DMSO treatment alone (*Figure 7C*). The effects of albendazole, however, were suppressed in FIP200 KO cells (*Figure 7D*). These findings imply that albendazole can degrade protein aggregates formed by a proteasome inhibitor in an autophagy-dependent manner.

## Lysosomal clustering is required for the degradation of αSyn aggregates

Next, we explored whether albendazole can degrade αSyn, the causative protein in PD. We established cellular models of PD wherein αSyn aggregation was prompted by the transduction of αSyn fibrils as seeds in αSyn-overexpressing cells (*Nonaka et al., 2010*). These αSyn fibrils were introduced into the SH-SY5Y cell line, which stably expressed the GFP-fused human αSyn. Notably, confocal microscopy analyses showed that upon the introduction of αSyn fibrils, lysosomes accumulated around the αSyn aggregates (*Figure 8A*). Albendazole further enhanced the concentration of lysosomes around these αSyn-GFP aggregates (*Figure 8B*). On electron microscopy, albendazole treatment induced a higher recruitment rate of autophagosomes and lysosomes to the αSyn aggregates than the control (*Figure 8C*). These observations suggest that lysosomes actively converge to aid in breaking down αSyn aggregates.

To verify the degradation of αSyn aggregates by albendazole, we introduced αSyn fibrils to induce αSyn aggregation, washed out the fibrils, and then treated cells with albendazole, either with or without bafilomycin A1. Subsequently, we assessed the αSyn concentration in the Triton-X insoluble fraction (*Figure 8D*). The introduction of αSyn fibrils notably increased αSyn-GFP levels, indicating αSyn aggregation in the Triton-X insoluble fraction. However, these levels decreased post-fibril washout after an 8 hr culture with standard media. This decline was mildly inhibited by bafilomycin A1, suggesting that basal autophagy degrades aggregated αSyn. Furthermore, albendazole treatment caused a marked tendency toward decreased αSyn-GFP levels compared with DMSO treatment. This decrease was also hindered by bafilomycin A1 (*Figure 8D and E*). This pattern indicates that albendazole mediates αSyn

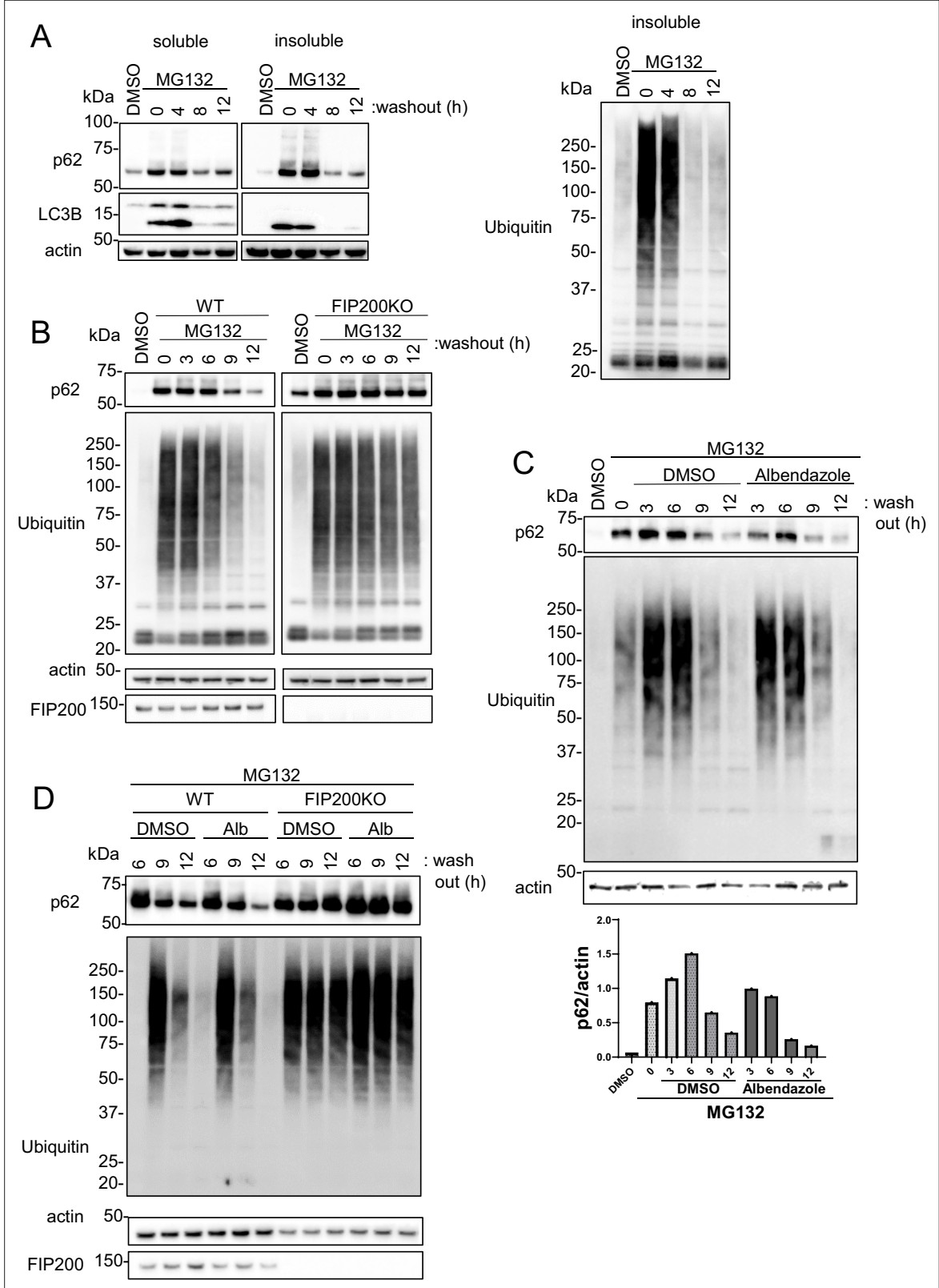

**Figure 7.** JIP4-dependent lysosomal clustering is involved in aggresome clearance by MG132. (**A**) SH-SY5Y cells were treated with MG132 (1 μM) for 16 hr to induce aggresome formation. After washing out MG132 with normal medium at intervals of 4, 8, or 12 hr, cell lysates were separated into Triton-X-100–soluble (soluble) and pellet fractions (insoluble), then subjected to sodium dodecyl sulfate-polyacrylamide gel electrophoresis and immunoblotting with the indicated antibody. (**B**) Wild-type (WT) or FIP200KO SH-SY5Y cells were treated with MG132 (1 μM) for 16 hr to induce

*Figure 7 continued on next page*

*Figure 7 continued*

aggresome formation. After washing out MG132 with normal medium at intervals of 3, 6, 9, or 12 hr, the same analysis as in panel A was performed. (**C**) SH-SY5Y cells were treated with MG132 (1 µM) for 16 hr. After washing out MG132 with either dimethyl sulfoxide (DMSO) or albendazole in normal medium at intervals of 3, 6, 9, or 12 hr, the same analysis as in panel A was performed. Bar graph showing the insoluble p62 ratio from panel C. (**D**) WT or FIP200KO SH-SY5Y cells were treated with MG132 (1 µM) for 16 hr to induce aggresome formation. After washing out MG132 with DMSO or albendazole in normal medium at intervals of 6, 9, or 12 hr, the same analysis as in panel A was performed.

The online version of this article includes the following source data for figure 7:

**Source data 1.** Uncropped blot images of Figure 7.

aggregate degradation through autophagy. Moreover, to evaluate the degradation of αSyn monomers by albendazole, we used SH-SY5Y cells stably expressing αSyn-Halo, to measure αSyn degradation by quantifying the cleaved Halo. Albendazole treatment induced a higher cleavage rate of Halo than DMSO treatment for 8 hr, suggesting that albendazole can degrade both αSyn monomers and aggregates (*Figure 8—figure supplement 1A*). The unchanged αSyn-GFP levels in the soluble fraction (*Figure 8D*) are likely due to the abundance of soluble αSyn-GFP. Therefore, albendazole enhances the autophagy-mediated degradation of αSyn-GFP aggregates and monomers.

Next, we investigated the degradation of αSyn-GFP aggregates by Torin1, which induces autophagy, but not lysosomal clustering. Torin1 also induced the degradation of aggregates, which was inhibited by bafilomycin A1 (*Figure 8—figure supplement 1B and C*). However, the degradation activity of albendazole was more vigorous than that of Torin1 (*Figure 8—figure supplement 1D and E*). Moreover, we found that Torin1 exhibited higher autophagic induction activity than albendazole (*Figure 6D*). These results suggest that albendazole, with its ability to gather lysosomes around the degradation substrate, more effectively facilitates degradation of insoluble αSyn than Torin1.

We have examined whether lysosomal clustering through the JIP4–TRPML1 pathway is crucial for the degradation of αSyn aggregates. We evaluated the degradation activity of insoluble αSyn induced by the addition of αSyn fibrils to αSyn-GFP cells under *JIP4, TMEM55B,* or *TRPML1* knockdown. Interestingly, the insoluble fraction assay showed that *JIP4* and *TRPML1* knockdown regulated the decrease of αSyn-GFP and p-αSyn levels in the insoluble fraction for both DMSO and albendazole treatments. The results were particularly more pronounced with *TRPML1* knockdown. However, the knockdown of *TMEM55B* did not produce such findings (*Figure 8—figure supplement 1F and G*). These data suggest that lysosomal clustering via the JIP4–TRPML1 pathway plays a significant role in αSyn degradation.

## Discussion

In this study, we developed a novel screening method for autophagy inducers that focused on lysosomal clustering and assessed their ability to degrade αSyn. We created a high-throughput screening system to quantify lysosomal accumulation by quantifying lysosomes located within a circular region centered on the MTOC. This was achieved through image analysis using INCellAnalyzer2200 and ImageJ (*Figure 1*). We discovered that topo-i and benzimidazole promoted lysosomal clustering via the JIP4–TRPML1 pathway (*Figures 2 and 4*). Specifically, topo-i induced lysosomal clustering through the phosphorylation of JIP4 at T217 (*Figure 5*). Compounds that stimulate lysosomal clustering also facilitate the fusion of autophagosomes and lysosomes near the MTOC by transporting autophagosomes via JIP4 (*Figure 6*). Furthermore, albendazole-induced lysosomal accumulation around αSyn aggregates surrounding the nucleus, subsequently degrading these aggregates in a lysosome-dependent manner (*Figure 8*). Our results indicate that lysosomal clustering plays a significant role in αSyn degradation. Thus, novel lysosomal-clustering agents have potential clinical applications through the targeting of αSyn aggregation in PD.

Topo-i and benzimidazole induce lysosomal clustering through distinct mechanisms. In our study, topo-i-driven lysosomal transport was regulated by JIP4 and TRPML1, but not by ALG2. This transport relies on the phosphorylation of JIP4 at T217 by CaMK2G. However, a previous study found that oxidative stress-induced lysosomal retrograde transport is influenced by the phosphorylated–JIP4–TRPML1–ALG2 complex (*Sasazawa et al., 2022*). Therefore, topo-i induces lysosomal clustering via a unique mechanism, involving p-JIP4 and TRPML1 without ALG2. Considering that TRPML1 acts as a reactive oxygen species sensor in lysosomes (*Zhang et al., 2016*), topo-i may induce ROS

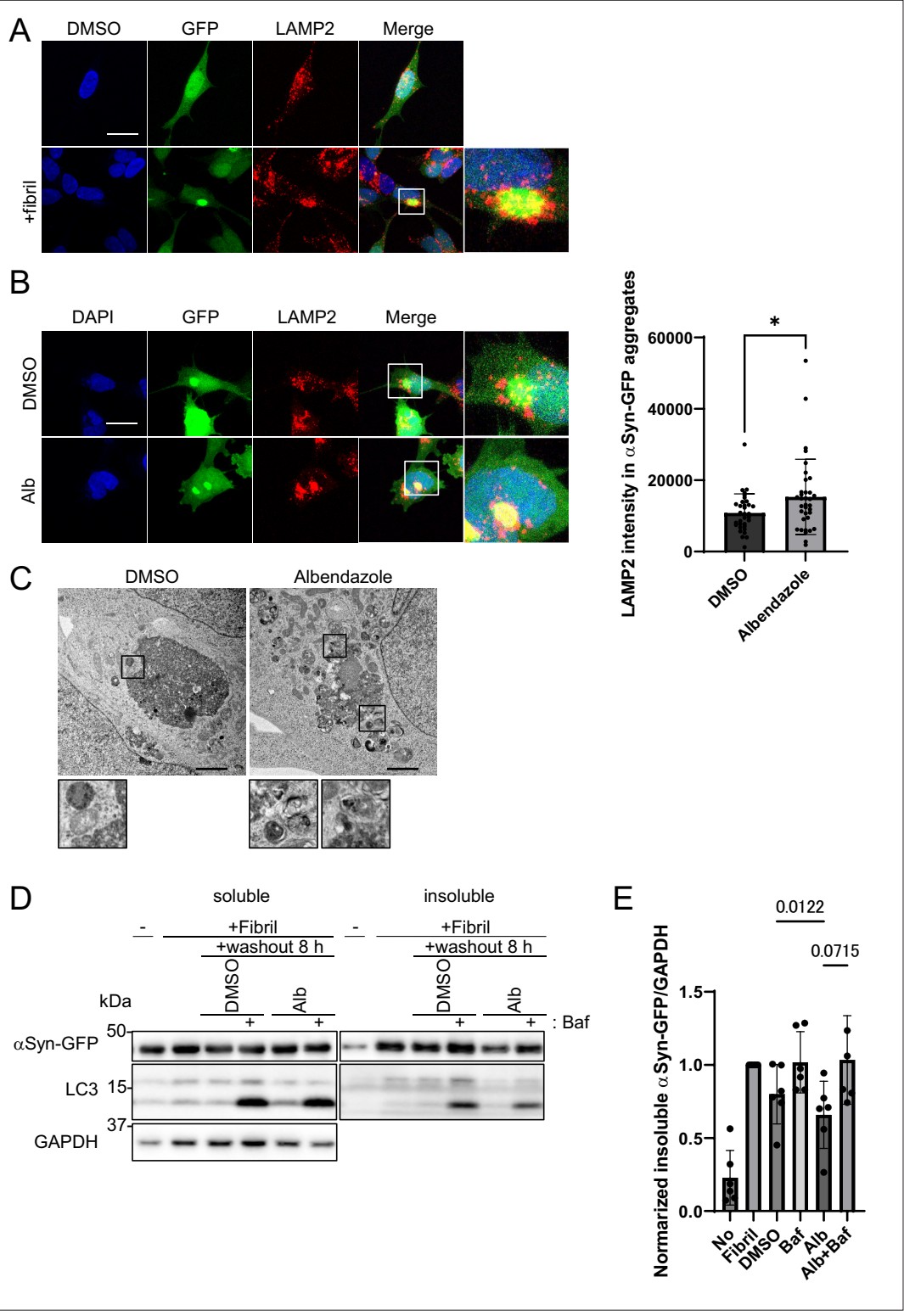

**Figure 8.** Albendazole reduces α-synuclein (αSyn)-green fluorescent protein (GFP) aggregates in a lysosome-dependent manner. (**A**) SH-SY5Y cells overexpressing αSyn-GFP were transfected with αSyn fibril (0.2 μg/mL) using Lipofectamine 3000. After 48 hr, the cells were fixed and stained with an anti-LAMP2 antibody (red) and 4',6-diamidino-2-phenylindole (DAPI; blue). These cells were imaged using a confocal microscope. Scale bar: 20 μm. (**B**) SH-SY5Y cells overexpressing αSyn-GFP were transfected with αSyn fibril (0.2 μg/mL) using Lipofectamine 3000. After 48 hr, the transfection reagent was washed out, and the SH-SY5Y cells were treated with dimethyl

*Figure 8 continued on next page*

*Figure 8 continued*

sulfoxide (DMSO) or albendazole (10 µM) with or without bafilomycin A1 (30 nM) for 8 hr. Cells were fixed and stained with an anti-LAMP2 antibody (red) and DAPI (blue). The cells were imaged using a confocal microscope. Scale bar: 20 µm. The colocalization of LAMP2 and αSyn-GFP aggregates was assessed by measuring the fluorescence values of lysosomes in contact within the αSyn-GFP aggregation area using ImageJ (n >30). * p<0.05, Wilcoxon test. (**C**) SH-SY5Y cells overexpressing αSyn-GFP were transfected with αSyn fibril (0.2 µg/mL) using Lipofectamine 3000. After 48 hr, the transfection reagent was washed out, and the SH-SY5Y cells were treated with DMSO or albendazole (10 µM), followed by electron microscopy analysis. Lysosome-like structures are enlarged at the bottom. Scale bar: 2 µm. (**D**) SH-SY5Y cells overexpressing αSyn-GFP were transfected with αSyn fibril (0.2 µg/mL) using Lipofectamine 3000. After 48 hr, the transfection reagent was washed out, and the SH-SY5Y cells were treated with DMSO or albendazole (10 µM) with or without bafilomycin A1 (30 nM) for 8 hr. Cell lysates were separated into Triton-X-100–soluble (soluble) and pellet fractions (insoluble), then subjected to sodium dodecyl sulfate polyacrylamide gel electrophoresis and immunoblotting with the indicated antibody. (**E**) Bar graph showing the insoluble αSyn-GFP ratio from panel D. Data are expressed as mean ± standard deviation. One-way analysis of variance and Dunnett's test.

The online version of this article includes the following source data and figure supplement(s) for figure 8:

**Source data 1.** Uncropped blot images of Figure 8D.

**Figure supplement 1.** Analysis of the degradation of α-synuclein (αSyn) aggregates.

production and stimulate TRPML1. Therefore, we assessed intracellular ROS in response to topo-i. Topo-i, such as teniposide, etoposide, and amsacrine, significantly increased ROS levels (***Figure 9A***). Moreover, N-acetyl-L-cysteine, a ROS scavenger, partially attenuated topo-i-induced lysosomal clustering (***Figure 9B***). Based on the activity of CAMK2G siRNA, as shown in ***Figure 5D and E***, and S5, topo-i may activate TRPML1 in a ROS-dependent manner and increase PI(3,5)P2 binding with TRPML1 (***Dong et al., 2010***). Consecutive Ca2+ release via TRPML1 activated CAMK2G and is followed by enhanced lysosomal transport toward the MTOC via JIP4 phosphorylation.

JIP4 serves as a scaffold protein. Specifically, it binds to p38 MAPK and facilitates its activation, undergoing phosphorylation in the process (***Kelkar et al., 2005***; ***Pinder et al., 2015***). Another study reported that curcumin-induced lysosomal clustering was inhibited using p38 inhibitors (***Willett et al., 2017***). Indeed, a p38 inhibitor prevented topo-i-induced lysosomal clustering, but it had no effect on benzimidazole (***Figure 9C and D***). Western blot analysis also showed that topo-i triggers p38 phosphorylation (***Figure 9E***). These data indicate the potential link of topo-i-induced lysosomal clustering' with p38. Since etoposide-induced DNA damage causes p38 phosphorylation (***Khedri et al., 2019***), topo-i-induced DNA damage could activate p38, facilitating lysosomal clustering. Moreover, topo-i failed to induce lysosomal clustering or autophagy flux in human adenocarcinoma HeLa cells (***Figure 10A***), suggesting that several molecules involved in lysosomal trafficking are absent in HeLa cells. In addition, autophagy flux assay for WB shows that topo-i did not induce autophagy in HeLa cells (***Figure 10B***). Further investigation is needed to elucidate the cellular specificity.

Benzimidazole-induced lysosomal clustering is suppressed by the knockdown of factors such as *JIP4*, *TRPML1*, *Rab7*, and *ALG2*, which indicates their potential involvement in the retrograde lysosomal transport of benzimidazole. Benzimidazole binds to β-tubulin, disrupting microtubule-based processes in parasites (***Lacey, 1988***). We observed that oxibendazole, even in low concentrations, promoted lysosomal clustering. However, at high concentrations, it disrupted tubulin production, thereby inhibiting lysosomal clustering (***Figure 10C***). We hypothesized that tubulin polymerization induced by benzimidazole plays a key role in inducing lysosomal clustering. To clarify this, we observed the behavior of tubulin filaments in response to various albendazole concentrations under confocal microscopy. Under conditions where albendazole was administered to induce lysosomal clustering, tubulin filaments were observed only near the MTOC, and the filaments in the cell periphery were disassembled. In contrast, when exposed to higher albendazole concentrations, tubulin filaments throughout the cell were disassembled, resulting in the inhibition of lysosomal clustering (***Figure 10D***). This would explain why benzimidazole exerts lysosomal clustering activity within a narrow concentration range.

Interestingly, benzimidazole not only failed to induce lysosomal clustering in HeLa cells, but it also transported lysosomes to the cell periphery. Moreover, this drug failed to induce autophagy flux (***Figure 10B***). Therefore, we examined the relationships between tubulin depolymerization and

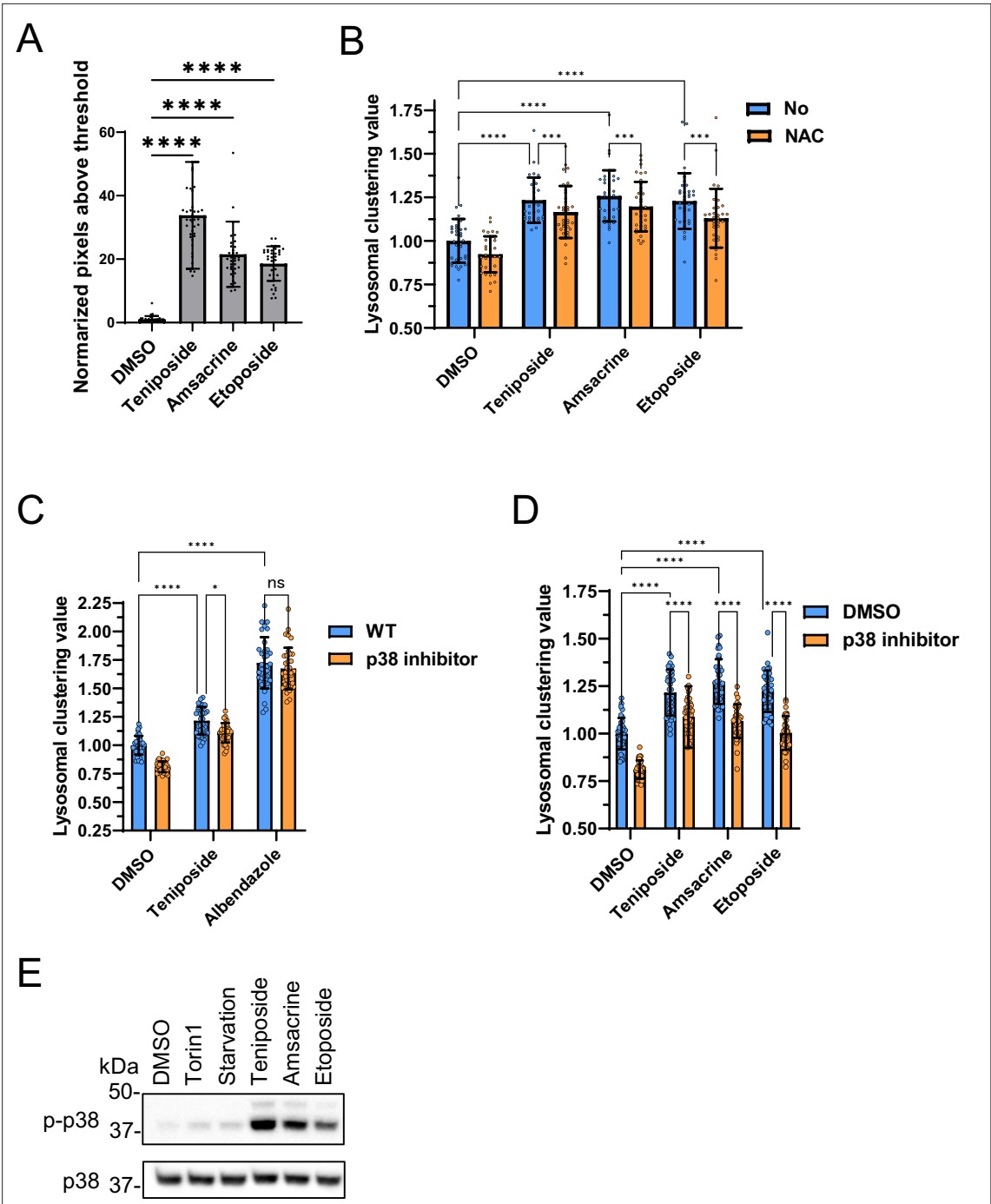

**Figure 9.** Analysis of lysosomal clustering mechanism of action for topoisomerase inhibitors. (**A**) SH-SY5Y cells were treated with the indicated compounds (10 μM) for 4 hr. The amount of intracellular reactive oxygen species (ROS) is examined by ROS Assay Kit -Highly Sensitive DCFH-DA (Dojindo) and the normalized pixels above threshold as measured using an INCellAnalyzer 2200 and ImageJ. (**B**) SH-SY5Y cell lines were pre-treated with 0.1 mM N-acetyl-L-cysteine for 24 hr and then treated with the indicated compound (10 μM) for an additional 4 hr. Cells were fixed and stained with anti-γ-tubulin (green) and anti-LAMP2 (red) antibodies. Lysosomal distribution was examined using an INCellAnalyzer 2200 and quantified using ImageJ software. (**C, D**) SH-SY5Y cells were cultured in 96-well black plates and treated with teniposide (10 μM), amsacrine (10 μM), etoposide (10 μM), or albendazole (10 μM) with or without the p38 inhibitor (SB203580). Following treatment, cells were fixed and stained with anti-γ-tubulin (green) and anti-LAMP2 (red) antibodies. They were then imaged using an INCellAnalyzer2200. INCellAnalyzer2200 images were analyzed for lysosomal clustering with ImageJ. The graph presents the lysosomal clustering value for either DMSO or the p38 inhibitor. (n>30, from three biological replicates). Data is shown as the mean ± standard deviation. ****p<0.0001, ***p<0.001, **p<0.01, *p<0.05, two-way analysis of variance and Tukey's test. N.S., not statistically

*Figure 9 continued on next page*

*Figure 9 continued*

significant. (**E**) SH-SY5Y cells were cultured and then treated with starvation medium, Torin1 (1 µM), teniposide (10 µM), amsacrine (10 µM), or etoposide (10 µM). Subsequently, cell lysates were immunoblotted with the indicated antibodies.

The online version of this article includes the following source data for figure 9:

**Source data 1.** Uncropped blot images of Figure 9D.

lysosomal clustering induced by albendazole in HeLa cells and found that albendazole did not induce lysosomal clustering but rather inhibited it at higher concentrations (*Figure 10A, D and E*). Interestingly, similar to SH-SY5Y cells, a low albendazole concentration (10 µM) induced tubulin depolymerization only at the cell periphery, whereas a high concentration (100 µM) depolymerized the entire cell (*Figure 10E*). However, unlike SH-SY5Y cells, no characteristic accumulation of tubulin filaments was observed near the MTOC under low albendazole concentration (10 µM); instead, they were arranged around the nucleus. Concurrently, lysosomes were around these dispersed tubulin filaments. Therefore, the differences in the effects of benzimidazole in HeLa and SH-SY5Y cells lie in the dose-dependent effects on the state of tubulin filaments.

Under *JIP4*, *TRPML1*, *ALG2,* and *Rab7* silencing, lysosomes may fail to interact with microtubules, resulting in the inhibition of lysosomal clustering. We postulated that albendazole-induced lysosomal clustering is not mediated by factors activated by specific stimuli in lysosomal transport but, rather, is induced by spatially constraining conventional lysosomal transport mediated by various adaptors (i.e. *JIP4*, *TRPML1*, *ALG2*, and *Rab7*) through tubulin disassembly.

Lysosomal clustering within cells has diverse roles. It is associated with the PD-related protein LRRK2 in JIP4-mediated lysosomal positioning (*Bonet-Ponce et al., 2020*; *Boecker et al., 2021*) and is involved in oxidative stress response (*Sasazawa et al., 2022*), pH regulation, and various cellular functions.

Lysosomal clustering promotes the fusion of autophagosomes with lysosomes and assists in breaking down aggregates and other cellular components. Previous studies reported that lysosomal retrograde transport enhances the fusion of autophagosomes and lysosomes during nutrient starvation, thus triggering autophagy (*Korolchuk et al., 2011*). Consistent with these findings, our study showed that in JIP4 KO cells, Halo-LC3 degradation was inhibited, implying the significant role of JIP4 in this process.

Ubiquitinated protein, p62 aggregation, and the proteasome itself, when blocked by the proteasome inhibitor MG132, are eventually degraded via the autophagy-lysosome pathway (*Jänen et al., 2010*; *Choi et al., 2020a*). Moreover, forcing lysosomes to the cell periphery by overexpressing Arl8b inhibits the breakdown of aggregates induced by MG132 (*Zaarur et al., 2014*). Our research provides the first evidence for the involvement of lysosomal clustering in the degradation of MG132-induced aggresomes and αSyn fibril transduction.

Notably, we showed that suppressing lysosomal clustering via the JIP4–pathway inhibits αSyn degradation. Furthermore, by inducing lysosomal clustering, albendazole can degradeαSyn aggregates more effectively than Torin1. Our results highlight the critical role of lysosomal clustering in enhancing protein aggregate degradation.

Aggregates formed by the mutant huntingtin protein associated with Huntington's disease are typically found both in the nucleus and the cytoplasm. These aggregates attract autophagosomes via the autophagy adapter CCT2, leading to their autophagy-mediated degradation (*Ma et al., 2022*). During this process, the recruitment of lysosomes to the aggregates is evident. In our studies, we also observed increased lysosomal clustering and elevated LC3-II levels in the insoluble fraction during αSyn aggregate formation (*Figure 8D*). This observation suggests that LC3-II binds to αSyn aggregates, and the subsequent accumulation of autophagosomes and lysosomes around the aggregates stimulates autophagy, leading to αSyn aggregate degradation. Receptor proteins may link LC3-II to αSyn and the degradation of huntingtin protein aggregates.

Overall, our results emphasize that lysosomal clustering is pivotal in aggregate degradation. Albendazole, by enhancing autophagy through lysosomal accumulation, is a promising therapeutic agent that can be administered to break down αSyn aggregates in Lewy body disease associated with PD.

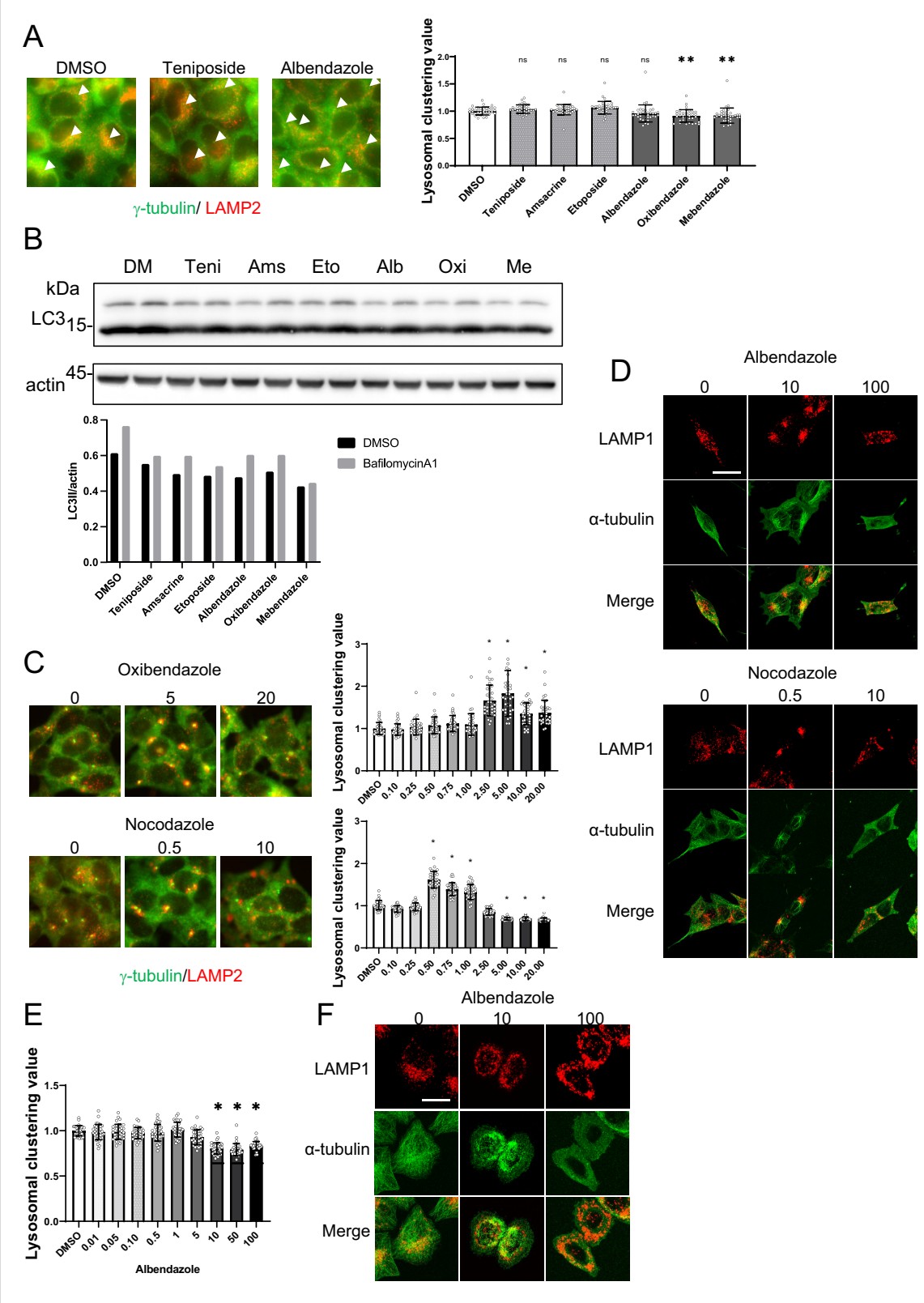

**Figure 10.** Analysis of lysosomal clustering mechanism of action for benzimidazole mechanism. (**A**) HeLa cells were treated with teniposide (10 µM), amsacrine (10 µM), etoposide (10 µM),albendazole (10 µM), oxibendazole (1 µM), or mebendazole (5 µM). Images were captured using an INCellAnalyzer2200. INCellAnalyzer2200 images were analyzed using ImageJ for lysosomal clustering. Scale bar: 20 µm.The graph presents the lysosomal clustering values (n>30, from three biological replicates). Data are expressed as mean ± standard deviation (SD). ****p<0.0001, ***p<0.001,

*Figure 10 continued on next page*

*Figure 10 continued*

**p<0.01, *p<0.05, two-way analysis of variance (ANOVA) and Tukey's test. N.S., not statistically significant. The experiment was technically replicated at least three times. (**B**) HeLa cells were treated with teniposide (10 µM), amsacrine (10 µM), etoposide (10 µM), albendazole (10 µM), oxibendazole (1 µM), or mebendazole (5 µM) for 4 hr. Cell lysates were immunoblotted with the indicated antibodies. The amount of LC3II was estimated using ImageJ software (bottom panel). (**C**) SH-SY5Y cells were cultured in 96-well black plates and treated with oxibendazole or nocodazole at specified concentrations (in µM). After treatment, cells were fixed and stained with anti-γ-tubulin (green) and anti-LAMP2 (red) antibodies, followed by imaging with an INCellAnalyzer2200. Scale bar: 20 µm. INCellAnalyzer2200 images were processed and analyzed using ImageJ for lysosomal clustering. The graph presents the lysosomal clustering values (n>30). Data are expressed as mean ± SD. *p<0.0001, as determined by one-way ANOVA and Dunnett's test (vs. dimethyl sulfoxide [DMSO]). (**D**) SH-SY5Y cells were treated with albendazole (10 and 100 µM) or nocodazole (0.5 and 10 µM) for 4 h. Cells were fixed and stained with LAMP1 (red) and α-tubulin (green) antibodies. Scale bar: 20 µm. (**E**) HeLa cells were treated with albendazole at specified concentrations (in µM). After treatment, cells were fixed and stained with anti-γ-tubulin (green) and anti-LAMP2 (red) antibodies, followed by imaging with an INCellAnalyzer2200. INCellAnalyzer2200 images were processed and analyzed using ImageJ for lysosomal clustering. The graph presents the lysosomal clustering values (n >30). Data are expressed as mean ± SD. *p<0.0001, as determined by one-way ANOVA and Dunnett's test (vs. DMSO). (**F**) HeLa cells were treated with albendazole (10 and 25 µM) for 4 hr. Cells were fixed and stained with LAMP1 (red) and α-tubulin (green) antibodies. Scale bar: 20 µm.

The online version of this article includes the following source data for figure 10:

**Source data 1.** Uncropped blot images of Figure 10B.

## Materials and methods

### Reagents

A chemical library consisting of about 1,200 clinically approved drugs in Japan was supplied by Prof. Hideyuki Saya (Fujita Medical University, Japan). Teniposide, amsacrine, etoposide, albendazole, oxibendazole, and mebendazole were purchased from Tokyo Chemical industry in Tokyo, Japan. Bafilomycin A1 was purchased from Sigma-Aldrich (St. Louis, MO, USA). Jak3 inhibitor VI was purchased from Merck Millipore (Burlington, MA, USA). SB203580 was purchased from AdipoGen Life Science (Liestal, Epalinges, Switzerland).

### Cell culture

SH-SY5Y cells (American Type Culture Collection, ATCC#CRL-2266, Virginia, USA, RRID:CVCL_0019) were cultured in Dulbecco's Modified Eagle's Medium (Nacalai Tesque, Kyoto, Japan) supplemented with 10% fetal bovine serum (FBS; MP bio, Ringmer, UK); 100 U/mL penicillin/streptomycin (Nacalai Tesque, Kyoto, Japan); minimum essential medium non-essential amino acid solution (Thermo Fisher Scientific, Waltham, MA, USA); 1 mM sodium pyruvate; and 2 mM L-glutamine at 37 °C with 5% $CO_2$. For starvation treatment, cells were washed with phosphate-buffered saline (PBS) and incubated in amino acid-free DMEM without serum (starvation medium; Wako). Tetracycline-on (Tet-on) cells were generated by lentiviral transduction with a pCW57.1 vector (Addgene plasmid 41393, David Root Lab) containing a single-vector Tet-on component and were cultured in the presence of 1 µg/mL doxycycline (Clontech, Mountain View, CA, USA) during induction. All the cell lines were tested negative for mycoplasma contamination (MycoAlert Detection Kit, Lonza, Basel, Switzerland).

### Stable cell line generation

Cells stably expressing LGP120-mCherry/GFP-γ-tubulin were established by transfecting the respective vectors into SH-SY5Y cells using Lipofectamine LTX (Thermo Fisher Scientific), followed by selection with G418 (Roche Diagnostics, Switzerland). mCherry- and GFP-positive cells were sorted using a Cell Sorter Aria (BD) and plated on a 96-well plate. SH-SY5Y cells stably expressing mCherry-GFP-LC3 and Halo-LC3 were generated by either lentiviral or retroviral transduction. HEK293 cells were transiently co-transfected with lentiviral vectors using PEI MAX reagent (Polysciences, Warrington, PA, USA). At 4 hr post-transfection, the medium was replaced with fresh culture medium. After 72 hr of culturing, the growth medium containing the lentivirus was collected. SH-SY5Y cells were then incubated with the virus-containing medium for 48 hr. Uninfected cells were removed using 1 µg/mL puromycin or 5 µg/mL blasticidin S (Wako).

### Lysosomal clustering analysis

Wild-type (WT) SH-SY5Y cells or those expressing GFP-γ-tubulin/LGP120-mCherry were cultured in 96-well black plates (Corning: 3603 or Greiner: 655892). After 48 hr of culture, cells were treated with

lysosome-clustering compounds for 4 hr. Cells were subsequently fixed with 4% paraformaldehyde (Nakarai Tesque) and stained with Hoechst33342 (Invitrogen, H3570) for 30 min. Image capture of cells was performed using a INCellAnalyzer2200 high-content imager (GE Healthcare, Chicago, USA). Lysosome distribution was quantified using Fiji software (ImageJ ver.2.1.0/1.53 c; *Schindelin et al., 2012*, RRID:SCR_002285). The image analysis was as follows:

I: The MTOC position is identified from the γ-tubulin image by ImageJ processing ('Apply-LUT'⇒'Subtract Background'⇒'Threshold').

II: A circle with a diameter of approximately 7 μm is centered on the MTOC coordinates.

III: This circle is superimposed on the LGP120 image, and the LGP120 fluorescence intensity within the circle is measured. The ratio of this fluorescence intensity to the whole-cell LAMP2 intensity is then calculated.

Processes I–III can be automatically performed using ImageJ programming. The lysosomal clustering value was defined as the ratio relative to the control.

## Immunoblotting

Western blot analysis was performed as previously described with minor modifications (*Sasazawa et al., 2012*; *Sasazawa et al., 2015*; *Sasazawa et al., 2022*). Cells were washed with cold PBS and lysed in lysis buffer (25 mM Tris–HCl pH 7.6, 150 mM NaCl, 1% NP-40, 1% sodium deoxycholate, 0.1% sodium dodecyl sulfate [SDS], and protease inhibitor cocktail) for 15 min on ice. The lysates were centrifuged at 20,000 g for 15 min to obtain soluble cell lysates. For SDS polyacrylamide gel electrophoresis (SDS-PAGE), samples were mixed with 4x SDS sample buffer and boiled at 95 °C for 5 min. Protein (10 μg) was added to each lane of the gel, separated by SDS-PAGE (Bio-Rad), and then transferred to a polyvinylidene difluoride (PVDF) membrane (Bio-Rad) using transfer buffer (25 mM Tris, 192 mM glycine, and 10% v/v methanol). Immunoblot analysis was performed using the indicated antibodies, and immunoreactive proteins were visualized using the West Dura Extended Duration Substrate (Thermo Fisher Scientific). The primary antibodies used were as follows: anti-LC3B (Cell Signaling Technology Inc, Danvers, MA, RRID:AB_2137707); anti-β-actin (Merck Millipore, RRID:AB_2223041); anti-phospho-mTORC1 (Ser2448; Cell Signaling Technology Inc, RRID:AB_330970); anti-mTORC1 (Cell Signaling Technology Inc RRID:AB_330978),;anti-phospho-ULK1 (Ser757; Cell Signaling Technology Inc RRID:AB_2665508); anti-ULK1 (Cell Signaling Technology Inc, RRID:AB_11178668); anti-phospho-p70S6K (Cell Signaling Technology Inc,RRID:AB_2269803); anti-p70S6K (Cell Signaling Technology Inc, RRID:AB_390722); anti-phospho-S6 (Cell Signaling Technology Inc, RRID:AB_916156); anti-S6 (Cell Signaling Technology Inc, RRID:AB_331355); anti-p62 (Abcam, RRID:AB_945626); anti-multi-ubiquitin (MBL, RRID:AB_592937); anti-FIP200 (D10D11; Cell Signaling Technology Inc, RRID:AB_2797913); anti-GAPDH (Cell Signaling Technology Inc, RRID:AB_10622025); anti-TMEM55B (Proteintech, RRID:AB_2879391); anti-ALG2 (R&D Systems, Inc, RRID:AB_10972311); anti-JIP4 (Abcam, RRID:AB_299021); and anti-GFP (Proteintech, RRID:AB_11182611).

To analyze phosphorylated JIP4 levels, lysates underwent 6% Phos-tag (50 μmol/L) acrylamide gel (FUJIFILM Wako Pure Chemical) electrophoresis. After electrophoresis, the gel was soaked in running buffer with 10 mM ethylenediaminetetraacetic acid (EDTA) twice for 10 min each, and then placed in transfer buffer for 10 min. Proteins were transferred to a PVDF membrane and probed with the anti-JIP4 antibody.

## Flow cytometry

Cells detached with trypsin-EDTA were resuspended in 10% FBS and 1 μg/mL 4',6-diamidino-2-phenylindole (DAPI) in PBS. They were then passed through a 70 μm cell strainer and analyzed using lasers equipped with NUV (BD) at 375 nm (DAPI), 488 nm (GFP), and 561 nm (RFP). Dead cells were identified by DAPI staining. For each sample, 10,000 cells were acquired, and RFP/GFP fluorescence ratios (red fluorescence intensity divided by green fluorescence intensity) were calculated for RFP-positive cells.

## Immunofluorescence

Cells were cultured on coverslips and fixed with 4% paraformaldehyde (Nakarai Tesque) for 30 min. For immunostaining, fixed cells were permeabilized with 50 μg/mL digitonin in PBS for 15 min, blocked with 4% bovine serum albumin in PBS for 30 min, and then incubated with primary and secondary

antibodies for 1 hr. Fluorescence images were captured using an LSM880 confocal laser scanning microscope (Carl Zeiss, Oberkochen, Germany). The primary antibodies used were as follows: anti-LAMP2 (Development Studies Hybridoma Bank, Iowa City, IA, UA; clone H4B4, RRID:AB_2134755); anti-γ-tubulin (Abcam, RRID:AB_2904198); anti-JIP4 (Thermo Fisher Scientific, RRID:AB_2642850).

## α-Synuclein aggregation assays

SH-SY5Y cells overexpressing α-synuclein (αSyn)-GFP were cultured for 24 hr. Cells were transfected with αSyn fibril (0.2 μg/mL; Cosmobio) using Lipofectamine 3000 (Thermo Fisher Scientific). At 48 hr post-transfection, cells were evaluated either by immunostaining or the insoluble fraction assay.

## Insoluble fraction assay

Insoluble fraction analysis was performed as previously described with minor modifications (*Oji et al., 2020*). Cultured cells were washed with ice-cold PBS and lysed in Triton-X-100 buffer (50 mM Tris, 150 mM NaCl, 1 mM EDTA, 1% Triton-X-100, 10% glycerol, and protease inhibitor cocktail [Thermo Fisher Scientific]). Lysates were then centrifuged at 100,000 $g$ for 20 min at 4 °C. The supernatants were designated as the detergent-soluble fraction. Pellets were washed with Triton-X-100 buffer, resuspended in SDS buffer (60 mM Tris-Cl [pH 6.8], 1 mM EDTA, 10% glycerol, 2% SDS, and protease inhibitor cocktail [Thermo Fisher Scientific]), and sonicated for 5 s×4 using a microtip sonicator. After centrifugation at 13,000 rpm for 15 min, the supernatants were identified as the detergent-insoluble fraction. Equal volumes of both insoluble and soluble fractions were boiled at 95 °C for 10 min in SDS sample buffer and then analyzed by immunoblotting.

## siRNA transfection

Transfection of SH-SY5Y cells with siRNAs was performed using Lipofectamine RNAiMax (Thermo Fisher Scientific) following the manufacturer's instructions. The siRNAs used were described previously (*Sasazawa et al., 2022*) as follows: TRPML1 (SASI_Hs01_00067195); ALG2 (SASI_Hs01_00030269); Rab7a (SASI_Hs01_00104360); JIP4 (SASI_Hs01_00194613); TMEM55B (SASI_Hs02_00322347); and RILP CaMK2G (SASI_Hs01_00118118), all obtained from Sigma-Aldrich. Non-coding siRNA was obtained from Dharmacon (Lafayette, CO, USA).

## Ultrastructural analysis of SH-SY5Y cells by electron microscopy

Cells were first treated with a solution containing 2% glutaraldehyde and 50 mM sucrose (Wako) in a 0.1 M phosphate buffer at pH 7.4. This was followed by post-fixation with 1% osmium tetroxide in the same buffer. The fixed samples were then progressively dehydrated using a graded ethanol series and embedded in Epok812 (Okenshoji). Ultrathin sections, approximately 70 nm thick, were prepared using a UC6 ultramicrotome (Leica) and subsequently stained with uranyl acetate and lead citrate. The sections were analyzed under a transmission electron microscope (JEM-1400, JEOL).

## Measurement of intracellular calcium

The quantitative analyses of intracellular calcium were performed using Fluo4-AM by a plate reader (BioTek Synergy H1, Agilent Technologies, California, USA). SH-SY5Y cells were cultured in 96-well black plates (Corning: 3603 or Greiner: 655892). After 48 hr of culture, cells were treated with lyso-somal clustering compounds for 4 hr. Cells were stained with Fluo4-AM (1 mg/ml) in BBS buffer (20 mM HEPES, 135 mM NaCl, 5.4 mM KCl, 2 mM CaCl2, 10 mM glucose) and washed once with BBS buffer before examination. With the excitation wavelength at 488 nm, fluorescence intensities at 518 nm were measured using the plate reader.

## The measurement of the reactive oxygen species

The amount of intracellular reactive oxygen species (ROS) is examined by ROS Assay Kit -Highly Sensitive DCFH-DA (Dojindo, Japan). SH-SY5Y cells were cultured in 96-well black plates (Corning: 3603 or Greiner: 655892). After 48 hr of culture, cells were treated with lysosomal-clustering compounds for 4 hr. Cells were subsequently treated with a working solution for 30 min, which was prepared by dilution of Highly Sensitive DCFH-DA Dye (1:1000) in reconstituted DMEM. After being washed with a working solution, phenol red-free DMEM (Thermo Fisher Scientific) were added and fluorescence

intensities were measured using an INCellAnalyzer 2200. The normalized pixels above the threshold of FITC were determined using ImageJ.

## Materials availability statement

The Fiji code used for data analysis is available from supporting files. All genetic reagents are available from the corresponding author upon reasonable request.

## Acknowledgements

We thank Motoki Date for the ImageJ macro programing.

This work was supported by JST SPRING (Grant Numbers JPMJSP2109 to Y.D.), JSPS KAKENHI (Grant Numbers 18K15464, 21K07425, 18KK0242, 18KT0027, and 22H02986 to S.S.).

## Additional information

### Funding

| Funder | Grant reference number | Author |
|---|---|---|
| Japan Society for the Promotion of Science London | 18K15464 21K07425 | Yukiko Sasazawa |
| Japan Society for the Promotion of Science London | 18KK0242 18KT0027 22H02986 | Shinji Saiki |

The funders had no role in study design, data collection and interpretation, or the decision to submit the work for publication.

### Author contributions

Yuki Date, Conceptualization, Resources, Data curation, Software, Formal analysis, Validation, Investigation, Visualization, Methodology, Writing – original draft, Writing – review and editing; Yukiko Sasazawa, Conceptualization, Resources, Supervision, Investigation, Methodology, Writing – original draft, Project administration, Writing – review and editing; Mitsuhiro Kitagawa, Resources, Supervision, Methodology, Writing – original draft, Writing – review and editing; Kentaro Gejima, Resources, Data curation, Software, Validation; Ayami Suzuki, Resources, Data curation; Hideyuki Saya, Conceptualization, Resources; Yasuyuki Kida, Resources, Supervision; Masaya Imoto, Conceptualization, Supervision; Eisuke Itakura, Conceptualization, Resources, Supervision; Nobutaka Hattori, Conceptualization, Resources, Supervision, Funding acquisition, Project administration, Writing – review and editing; Shinji Saiki, Conceptualization, Resources, Supervision, Funding acquisition, Investigation, Methodology, Writing – original draft, Project administration, Writing – review and editing

### Author ORCIDs

Yuki Date ⓘ https://orcid.org/0009-0004-1099-1829
Yukiko Sasazawa ⓘ https://orcid.org/0000-0002-0287-273X
Nobutaka Hattori ⓘ https://orcid.org/0000-0003-2305-301X
Shinji Saiki ⓘ https://orcid.org/0000-0002-9732-8488

### Decision letter and Author response

Decision letter https://doi.org/10.7554/eLife.98649.sa1
Author response https://doi.org/10.7554/eLife.98649.sa2

## Additional files

### Supplementary files
• MDAR checklist

**Data availability**

All data generated or analysed during this study are included in the manuscript and supporting files.

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
