## [Editor Report]

This study reports a valuable finding regarding the therapeutic efficacy of compounds fostering lysosomal clustering to enhance the clearance of protein aggregates in neurodegenerative disorders. The data were collected and analyzed using solid and validated methodology. While a deeper mechanistic understanding would have strengthened the study, the work will be of interest to cell biologist work on autophagy and lysosome and medical biologists working on neurodegenerative diseases.

---

## [Decision Letter]

[Editors' note: this paper was reviewed by Review Commons.]

---

## [Author Response]

General Statements [optional]

Thank you for the review of our paper entitled “Identification of novel autophagy inducers by accelerating lysosomal clustering against Parkinson's disease” (RC-2023-02232). We have carefully read the critiques and planed experiments. Below we include point-by-point responses to the questions raised by the reviewers. We believe this revision plans appropriately addresses the issues raised by Reviewers. Finally, all the authors would like to thank again the Editor and Reviewers for improving our manuscript by providing their invaluable comments and suggestions.

Point-by-point description of the revisionsReviewer #1 (Evidence, reproducibility and clarity (Required)):The manuscript by Date et al. employed a cell model by stably expressing LGP120-mCherry and GFP-γ-tubulin to carry out high-content screening in search of chemical compounds that enhance lysosomal clustering and autophagy. They found 6 clinically approved drugs categorized as topoisomerase II inhibitors and the benzimidazole class. They further validated these compounds by a set of well-designed experiments including autophagy flux assays and mTOR dependence. In the mechanistic study, they demonstrated the compounds induce lysosomal clustering in a JIP4-TRPML1-dependent manner. In a PD cell model, one of the compounds albendazole exhibited the effect on boosting the degradation of insoluble α-synuclein. The study is of interest, and the cell model and the approach generated by the authors would be transferable for future studies of other high-content imaging screening. Most of the data is clear and convincing.Major comment1) In addition to its role in facilitating a-syn turnover by autophagy, Is the chemical protective against a-syn toxicity?

As suggested by the Reviewer, we examined the cytotoxicity of αSyn aggregates in SH-SY5Y cells overexpressing αSyn-GFP by LDH assay. As shown in the revised version of Figure 1, αSyn aggregates induced by introducing αSyn fibrils into SH-SY5Y cells overexpressing αSyn-GFP did not exhibit any cytotoxicity. In addition, we observed no significant change in cell death after 8 hours of treatment with albendazole compared with DMSO.

Previous studies have reported that induced pluripotent stem cells (iPSCs) derived from patients with PD with a triplication of the human *SNCA* genomic locus exhibited reduced capacity for differentiation into dopaminergic or GABAergic neurons, decreased neurite outgrowth, and lower neuronal activity compared with control cultures, albeit without showing cytotoxicity (*Cell Death and Disease* 6: e1994, Oliveira et al., 2015). Given this context, we were thus unable to conduct the suggested assessment due to technical limitations. Therefore, we consider the evaluation of the recovery of αSyn toxicity by drug treatment challenging in this cellular model using fibril αSyn.

2) Please elaborate why albendazole does not change the levels of soluble a-syn, but those of insoluble, as shown Figure 8D.

The unchanged αSyn-GFP levels in the soluble fraction (Figure 8D) are likely due to the abundance of soluble αSyn-GFP. To evaluate the autophagic degradation of αSyn monomers, we used SH-SY5Y cells stably expressing αSyn-Halo and measured αSyn degradation by quantifying cleaved Halo. As shown in the revised version of Figure 2, albendazole treatment induced a higher cleavage rate of Halo than DMSO treatment for 8 h, suggesting that albendazole degrades both αSyn monomers and αSyn aggregates. We have added the data in Figure 8—figure supplement 1A, and the description of these experiments in the Results section (page 10, lines 362 to 367).

3) Figure 6A shows that some of the compounds (Teniposide, Amsacrine) affect the levels of JIP4. Can albendazole also reduce JIP4 levels. It might be interesting to test this, as JIP4 is important for lysosomal clustering.

As the Reviewer pointed out, JIP4 is essential for lysosome accumulation. However, our data showed decreased JIP4 levels with the addition of lysosomal-clustering compounds. We hypothesized that this response was caused by the autophagy-induced degradation of JIP4. The decrease in JIP4 levels was detected by western blot after 4 h of treatment with 10 μM of teniposide. Moreover, the decrease in JIP4 levels induced by teniposide was suppressed by co-treatment with bafilomycin A1, indicating that JIP4 was degraded by teniposide-induced autophagy, as shown in the revised version of Figure 3. We have added the data in Figure 6—figure supplement 1 and the related description of these experiments in the Results section (page 8, lines 289 to 296).

Minor comments:The writing is good generally. Please tide up the text in a few occasions to make the expressions more formal.

We have revised our manuscript to adopt a more formal tone.

Reviewer #1 (Significance (Required)):Significance: The study generated a new approach for high-throughput screening of compounds to enhance lysosomal clustering.Audience: Basic and clinical researchExpertise: Programmed cell death, neurodegenerative diseasesReviewer #2 (Evidence, reproducibility and clarity (Required)):In this study, the authors focused on lysosome positioning and autophagy activity to search for novel agents effective against Parkinson's disease. As a result, several compounds were successively identified, including Topoisomerase inhibitors and Benzimidazole. Authors showed that these agents regulate lysosomal positioning through different pathways but commonly require JIP4 to regulate lysosomal positioning and subsequent autophagy. They also showed that albendazole treatment promoted the degradation of insoluble ubiquitinated proteins and αSyn in cultured cells.Major Comments.1) Two compounds, for instance teniposide and albendazole both requires JIP4 and/or TRPML1 to regulate lysosomal positioning and autophagy but their action seems different. What is the actual mechanism by which these compounds require JIP4/TRPML1. How inhibition of Topoisomerase leads to increase of JIP4 phosphorylation? Do teniposide and albendazole both affect calcium release from TRPML1?

We previously reported that acrolein/H2O2 accelerates lysosomal retrograde trafficking by TRPML1 and phosphorylated JIP4. Mechanistically, JIP4 was phosphorylated by CaMK2G activated by ca^2+^ released from TRPML1 (*EMBO J* 41: e111476, Sasazawa et al., 2022). TRPML1 acts as a reactive oxygen species (ROS) sensor in lysosomes (*Nat Commun* 7: 12109, Zhang et al., 2016). We concluded that acrolein induces ROS production, which then activates TRPML1. (*EMBO J* 41: e111476, Sasazawa et al., 2022). Therefore, topoisomerase inhibitors (topo-i) may induce ROS and stimulate TRPML1. We examined intracellular ROS levels in response to topo-i. As shown in revised Figure 4A, the topo-i teniposide, etoposide, and amsacrine significantly increased ROS levels. Moreover, N-acetyl-L-cysteine, an ROS scavenger, partially attenuated lysosomal clustering induced by topo-i (revised Figure 4B). In addition, ca^2+^ imaging showed that teniposide, but not albendazole, upregulates ca^2+^ flux (revised Figure 4C). Based on the activity of CaMK2G siRNA as shown in Figure 5D, 5E, and Figure 5—figure supplement 1, topo-i may activate TRPML1 in a ROS-dependent manner and increase PI(3,5)P2 binding with TRPML1 *(Nat Commun* 1, 38, Dong et al., 2010). Consecutive ca^2+^ release via TRPML1 activated CaMK2G and is followed by enhanced lysosomal transport toward the MTOC via JIP4 phosphorylation.

We have added the revised Figure 4A and 4B data in Discussion—figure supplement 1A and 1B, and the related description of these experiments in the Discussion section (page 11, lines 406 to 415). We have also added the data in revised Figure 4C to Figure 5—figure supplement 1 and the related description of these experiments in the Results section (page 8, lines 269 to 271).

Conversely, we showed that benzimidazoles, including albendazole, induce lysosomal clustering mediated by JIP4, TRPML1, ALG2, and Rab7. Moreover, benzimidazoles showed lysosomal clustering activity within a narrow concentration range, as shown in Discussion—figure supplement 2C. Benzimidazoles inhibit tubulin polymerization (*Int J Paras* 18:885–936. Lacey et al., 1988). We hypothesized that the effect of tubulin polymerization induced by benzimidazole plays a key role in the induction of lysosomal clustering as described in the Discussion section. To clarify this, we observed the behavior of tubulin filaments in response to various albendazole concentrations under confocal microscopy. As shown in revised Figure 4D, conditions where albendazole was administered to induce lysosomal clustering, tubulin filaments were observed only near the MTOC, and the filaments in the cell periphery were disassembled. In contrast, when exposed to higher albendazole concentrations, tubulin filaments throughout the cell were disassembled, resulting in the inhibition of lysosomal clustering This would explain why benzimidazole exerts lysosomal clustering activity within a narrow concentration range. Under *JIP4*, *TRPML1*, *ALG2* and *Rab7* silencing, lysosomes may fail to interact with microtubules, resulting in the inhibition of lysosomal clustering. We postulated that albendazole-induced lysosomal clustering is not mediated by factors activated by specific stimuli in lysosomal transport but, rather, is induced by spatially constraining conventional lysosomal transport mediated by various adaptors (i.e., JIP4, TRPML1, ALG2, and Rab7) through tubulin disassembly. We have added the data in Discussion—figure supplement 2C and D and the related description of these experiments in the Discussion section (page 12, lines 435 to 443).

2) The authors should clarify the functional advantage of these drugs identified in this study as drugs for Parkinson's disease by comparing with known autophagy inducers such as Torin1 or rapamycin.

To evaluate the functional advantage of lysosome-clustering compounds over Torin1, we evaluated the degradation activity of insoluble αSyn induced by the addition of αSyn fibrils to αSyn-GFP cells. Torin1 induced the degradation of insoluble αSyn by autophagy, as shown in revised Figure 5A. However, the degradation activity of albendazole was more vigorous, as shown in revised Figure 5B. In contrast, we observed that Torin1 exhibited more autophagic induction activity than albendazole, as assessed using Halo-LC3. Similar results were obtained with teniposide (revised Figure 5C). These results suggest that albendazole, with its ability to concentrate lysosomes around the degradation substrate, facilitates more effective degradation of insoluble αSyn than Torin1. This presents a significant advantage in the development of therapeutics for Parkinson's Disease. Moreover, Torin1 acts on the upstream signals of autophagy by inhibiting mTORC1, potentially impacting diverse cellular responses. Conversely, compounds that induce lysosomal clustering target the final step of autophagic degradation, which may have fewer side effects. We have added the description of these experiments in the Results section (page 10, lines 369 to 375) and the Discussion section (page 13 lines 479 to 481) and presented the data in Figure 8—figure supplement 1B－E and Figure 6D.

3) Related to the previous question, in Figure 6A and B additional data comparing novel compounds with established autophagy inducers, such as torin1 and rapamycin, should be included and discussed.

As indicated in a previous response, we evaluated the autophagic induction activity of Torin1, and the results have been added to Figure 6D. In addition, co-treatment with Torin1 and teniposide or albendazole induced autophagy more effectively than Torin1 treatment alone, without affecting mTOR inhibition activity (revised Figure 4C). These findings indicate that the induction of autophagy by lysosomal clustering compounds is not caused by autophagosome formation but by the formation of autolysosomes. We have added a description of these experiments in the Results section (page 9, lines 319 to 325) and have added the data in Figure 6D.

4) The authors should examined whether increased degradation of insoluble proteins and αSyn are dependent on JIP4.

As the Reviewer suggested, we have examined whether lysosomal accumulation through the JIP4-TRPML1 pathway is crucial for the degradation of αSyn aggregates. We evaluated the degradation activity of insoluble αSyn induced by the addition of αSyn fibrils to αSyn-GFP cells when JIP4, TMEM55B, or TRPML1 were knocked down. Interestingly, the insoluble fraction assay showed that JIP4 and TRPML1 knockdown regulated the decrease of αSyn-GFP and p-αSyn levels in the insoluble fraction for both DMSO and albendazole treatments. The results were particularly more pronounced with TRPML1 knockdown. However, the knockdown of TMEM55B did not produce such findings (revised Figure 6). These data suggest that lysosomal clustering via the JIP4–TRPML1 pathway plays a significant role in αSyn degradation. We have added a relevant description in the Results section (page 10, lines 376 to 380) and have added the data in Figure 8—figure supplement 1F and 1G.

5) Authors only utilized. SH-SY5Y cells in this study. It is important to examine whether these compounds also regulate lysosomal positioning and autophagy in other cell lines.

As per the Reviewer’s suggestion, we evaluated the lysosomal-clustering activity induced by topo-i and benzimidazole in human adenocarcinoma HeLa cells. As shown in revised Figure 7A and 7B compounds do not induce lysosomal clustering or autophagy in HeLa cells. Furthermore, in the case of benzimidazole, they transport lysosomes to the cell periphery. Previously, we found that oxidative stress accumulates lysosomes in a neuroblastoma-specific manner through the TRPML1–phosphoJIP4-dependent mechanism (*EMBO J* 41: e111476, Sasazawa et al., 2022). Since we have demonstrated that topo-i-mediated lysosomal trafficking is dependent on the TRPML1–phosphorylated JIP4 complex, we hypothesized that several molecules involved in lysosomal trafficking are absent in HeLa cells.

In contrast, we showed that albendazole-induced lysosomal clustering is due to tubulin depolymerization. Therefore, we examined the relationships between tubulin depolymerization and lysosomal clustering induced by albendazole in HeLa cells and found that albendazole did not induce lysosomal clustering but rather inhibited it at higher concentrations (revised Figure 7B). Interestingly, similar to SH-SY5Y cells, a low albendazole concentration (10 μM) induced tubulin depolymerization only at the cell periphery, whereas a high concentration (100 μM) depolymerized the entire cell (revised Figure 7C). However, unlike SH-SY5Y cells, no characteristic accumulation of tubulin filaments was observed near the MTOC under low albendazole concentration (10 µM); instead, they were arranged around the nucleus. Concurrently, lysosomes were around these dispersed tubulin filaments. Therefore, the differences in the effects of benzimidazole in HeLa and SH-SY5Y cells lies in the dose-dependent effects on the state of tubulin filaments. We have added a relevant description in the Discussions section (page 12, lines 425 to 429, and 444 to 461).

6) The authors conclude that the six compounds do not mediate mTOR signaling in Figure 3, but should more carefully describe in the manuscript why they performed this experiment and what the results mean for.

As per the Reviewer’s advice, we have changed the description in the manuscript as follows:

Previous studies have shown that lysosomal retrograde transport regulates autophagic flux by facilitating autophagosome formation by suppressing mTORC1 and expediting fusion between autophagosomes and lysosomes (Kimura et al., 2008; Korolchuk et al., 2011). Conversely, we recent found that acrolein/H2O2 induces lysosomal clustering in an mTOR-independent manner (Sasazawa et al., 2022). In this study, we aimed to identify pharmacologic agents that act downstream rather than upstream in the autophagy pathway, with the goal of minimizing side effects. Therefore, we evaluated the effects of the compounds on the mTOR pathway. As shown in Figure 3, these compounds induced lysosomal clustering without affecting mTOR activity, indicating their potential as promising candidates for PD therapy. We have added the description of these experiments in the Results section (page 6, lines 203 to 209 and line 218 to 219).

Minor comments.1) The name of the compound should be written in the red point of Figure 2A.

We have included the names of the six compounds identified and are listed in Figure 2A.

2) Regarding images of Figure 2B, the magnified images and quantitative data should be added.

We have included magnified images, as well as the quantitative results of lysosome clustering analysis using INCellAnalyzer2200 in Figure 2B.

3) The results of Figure 2C need to be explained more carefully. A quantitative data is missing.

We have included the quantitative results of western blot in Figure 2B.

4) Figure S2, which compares autophagy activity with conventional agents, should be quantified and added to the Figure 3.

We have presented the results of RFP/GFP quantification performed by FACS analysis using SH-SY5Y cells stably expressing RFP-GFP-LC3 in Figure 2—figure supplement 2, which is equivalent to the quantification of the data in the Figure 2—figure supplement. These data are now presented as Figure 2—figure supplement 2B. Since Figure 3 focuses on mTOR signaling, we preferred to retain the figure number.

5) In the statistical analysis of Figure 4B, the clustering value was increased by siRILP, which should be briefly described in the manuscript.

On the contrary, the enhancement of lysosomal retrograde transport in *RILP* knockdown cells in Figure 4B suggests the potential involvement of RILP in anterograde transport. However, to the best of our knowledge, no reports have investigated this matter. We presume that negative feedback mechanisms may be present. We have added this description to the Results section (page 7 lines 240 to 243).

6) In Figure 4A and B, it is possible that the knockdown efficiency of siRILP and siTMEM55B was not sufficient to observe the effect on lysosomes, and this concern should be described in the manuscript.

We established starvation conditions, which induce TMEM55B-dependent lysosomal retrograde transport, as a positive control and evaluated the lysosomal induction activity of compounds when TMEM55B was knocked down. As shown below, lysosome accumulation was suppressed only when subjected to starvation treatment, indicating sufficient knockdown efficiency of TMEM55B. These compounds induced lysosomal clustering independently of TMEM55B, unlike under starvation conditions. We have added a description of these experiments in the Results section and presented the data in Figure 4—figure supplement 2A (page 7, lines 234 to 239).

On the other hand, we were unable to establish a positive control for RILP knockdown experiments because conditions that regulate RILP-dependent lysosomal distribution dependent are not understood. While we cannot completely rule out the possibility of insufficient knockdown efficiency, considering that RILP knockdown appears to paradoxically enhance lysosomal induction, as mentioned above, it is reasonable to assume that the knockdown effect has occurred.

7) The authors should add the results of the WB experiment showing the amount of JIP4 protein in Figure 5G.

We have added western blot data that introduce flag-JIP4 into JIP4KO SH-SY5Y cells, which are presented in Figure 5G.

8) In Figure 5F, images of JIP4KO cells that do not express FLAG-JIP4 should be added as controls, and further quantification should be done on cells in all three conditions.

We have added immunofluorescence data that do not express flag-JIP4 in Figure 5F, which had been obtained simultaneously during the acquisition of other images. Furthermore, we quantified lysosomal distribution, which is shown in Figure 5E. Using ImageJ, we automatically delineated approximately 70% of the cell area toward the cell center and designated the region excluded from this area as the cellular peripheral region (revised Figure 9A). Subsequently, we quantified the proportion of lysosomes contained within that region in cells expressing flag-JIP4 (revised Figure 9B). We have added this experimental data in Figure 5E.

9) In Figure 6A, the total amount of JIP4 seems to change in some agent treatments, which needs to be explained.

As per our response to Reviewer 1, we evaluated the decrease in JIP4 expression by WB after 4 h of treatment with 10 μM teniposide. The teniposide-induced decrease of JIP4 was suppressed by bafilomycinA1 co-treatment, indicating that JIP4 was degraded by teniposide-induced autophagy (revised Figure 3). We have added the data in Figure 6-igure supplement 1, and the related description of these experiments have been added to the Results section (page 8, lines 292 to 296).

10) In Figure 7C and D, the effect of drug treatment on the amount of ubiquitinated proteins should also be checked.

We have included ubiquitin protein blots in Figure 7C and 7D.

11) In Figure 8B, it is described that lysosomes are more localized in αSyn by drug treatment, but more convincing images and quantitative data are needed.

The colocalization of LAMP2 and αSyn-GFP aggregates was assessed by measuring the fluorescence values of lysosomes in contact within the αSyn-GFP aggregation area using ImageJ. We have added this quantified data in Figure 8D.

Reviewer #2 (Significance (Required)):Although the reviewer appreciates the discovery of novel drugs to induce autophagy through regulating lysosomal positioning, the detailed action of these compounds and their superiority in the field are not clear.Reviewer #3 (Evidence, reproducibility and clarity (Required)):In this manuscript, Date et al. sought to identify compounds that promote protein aggregates clearance – in particular those formed by mutant α synuclein. Briefly the authors screened a library of clinically approved compounds for inducers of lysosomal clustering followed by a secondary screen for autophagy inducers. By this two-step procedure, the authors identified three topoisomerase inhibitors and three anthelmintics as hits. Next, the authors unveiled that lysosomal clustering induced by these compounds is independent of mTORC1 but requires TRPML1 and JIP4. Moreover, the topoisomerase inhibitors hits involved phosphorylation of JIP4 while the anthelmintics additionally required Rab7 and ALG2. Intriguingly, the authors found that lysosomal clustering was prerequisite to autophagy induction. Focusing on the class of anthelmintics (i.e. albendazole) the authors showed that these induce autophagy to degrade aggregates formed upon proteasome inhibition. Lastly, the authors demonstrated that albendazole also led to increased degradation of αSyn aggregates through autophagy induction.Major points1) Most importantly, the authors need to tone down the significance of their findings throughout the manuscript. For examples, they should restrain from using "nullified" when it is really reduced only by 10-25 %.

We have changed the description in the manuscript according to the Reviewer’s suggestion.

2) The authors claim that the topoisomerase inhibitors led to JIP4 phosphorylation while Figure 5C actually shows the opposite (partially reduced phosphorylation compared to DMSO treatment) and the Jak3 inhibitor has no obvious effect. The authors should quantify the phostag results.

We agree with the Reviewer that the Phos-tag PAGE results of JIP4 in Figure 5C is complicated, and the bands were not clear. We have replaced these with more robust data (Figure 5C).

3) Figure 6A/B: Why do all compounds except Mebendazole affect the abundance of JIP4?

As per our response to Reviewer 1, we evaluated the decrease in JIP4 expression by WB after 4 h of treatment with 10 μM teniposide. The teniposide-induced decrease of JIP4 was suppressed by bafilomycinA1 co-treatment, indicating that JIP4 was degraded by teniposide-induced autophagy (revised Figure 3). We have added the data in Figure 6—figure supplement 1, and the related description of these experiments have been added to the Results section (page 8, lines 292 to 296).

4) Figure 7C: The blot is not convincing. The authors should quantify this effect.

We evaluated and confirmed the degradation of p62 by albendazole, as shown in Figure 7C.

Reviewer #3 (Significance (Required)):Overall, the work of Date and colleague highlights the role of lysosomal clustering in clearing protein aggregates. Importantly, the identified classes of compounds might open new avenues for rationalizing treatment strategies for neurodegenerative diseases. However, several critical points remain.